## RESEARCH ARTICLE

# Enhanced RNA quality control maintains long-term regenerative ability in planarians

Michael Zelko[1],[*],[§], Danyan Li[1],[‡],[§], Sudheesh Allikka Parambil[1], Axel Poulet[1], Andrew Verdesca[1], Kaspar Mazeika[1], Krishnakali Dasgupta[1] and Josien C. van Wolfswinkel[1],[2],[3],[4],[¶]

## ABSTRACT

Planarians have proficient regenerative abilities that persist undiminished throughout adulthood, mediated by their stem cells (neoblasts). It is unclear how planarians accomplish this, as most animals show age-related declines in health and regeneration. Neoblasts express the conserved RNA regulatory PIWI protein SMEDWI-1, homologs of which are found in germ cells and long-lived cells in other systems. We previously found that loss of SMEDWI-1 from the neoblasts results in accumulation of non-coding and aberrant RNAs. Here, we report that, over time, SMEDWI-1-depleted animals develop defects in wound repair and regeneration, alterations in secreted proteins, and increased intracellular protein aggregation. Our data indicate that these defects result from misassembly of the signal recognition particle (SRP), a ribonucleoprotein (RNP) responsible for co-translational protein secretion that contains a non-coding RNA as a scaffold. In the absence of tight regulation of non-coding RNA, as provided by SMEDWI-1, gradual accumulation of RNAs leads to imbalances in essential cellular machinery such as the SRP, resulting in compromised proteostasis and progressive loss of organismal vigor.

KEY WORDS: Regeneration, RNA control, PIWI protein, Non-coding RNA, Proteostasis, Aging

## INTRODUCTION

Wound repair and regeneration are crucial processes in organismal life. At the end of embryonic development, the main structures that make up the adult organism have been formed, but developmental processes continue to play an essential role in body maintenance. The cells and structures of an organism inevitably sustain damages over time, both from internal and external origins, resulting in diminished tissue health and increased organismal fragility. To maintain function and ultimately viability, lesions need to be repaired, cells need to be replaced and structures need to be rebuilt.

[1]Department of Molecular Cellular and Developmental Biology, Yale University, New Haven, CT 06511, USA. [2]Yale Stem Cell Center, Yale School of Medicine, New Haven, CT 06511, USA. [3]Yale Center for RNA Science and Medicine, Yale School of Medicine, New Haven, CT 06511, USA. [4]Yale Center for Aging Research (Y-AGE), Yale School of Medicine, New Haven, CT 06511, USA.
[*]Present address: Cell Molecular Biology Graduate Program, University of Chicago, Chicago, IL 60637, USA. [‡]Present address: Department of Basic Medical Sciences, School of Medicine, Tsinghua University, Beijing, China.
[§]These authors contributed equally to this work

[¶]Author for correspondence ( josien.van.wolfswinkel@yale.edu)

J.C.v.W., 0000-0003-4221-3218

Wound healing and regeneration thus are crucial to maintain organismal integrity and extend organismal life.

While injuries and damage accumulate over time, organismal regenerative abilities typically decrease with age, resulting in a progressive decline in organismal health, commonly known as aging. Remarkably, some animals, such as the planarian *Schmidtea mediterranea*, appear able to maintain organismal health indefinitely (Deere et al., 2024; Gambino et al., 2020; Mouton et al., 2011; Oviedo et al., 2008; Perrigue et al., 2015; Sahu et al., 2017). *S. mediterranea* has an undetermined lifespan, and does not show a decrease in regenerative ability or wound repair with age. This is in large part ascribed to the presence of a population of adult stem cells called neoblasts, which allow the animals to continuously replace any aged or damaged cells with new specimens (Reddien and Sanchez Alvarado, 2004). This implies that the neoblasts of *S. mediterranea* need to maintain their functional integrity for extended periods of time. In the asexual strain of *S. mediterranea*, which does not generate mature germ cells and thus always remains in its adult stage, the neoblast population has sustained the animals for millions of years, without detectable decline (Lázaro et al., 2011), suggesting that the degenerative processes that plague other cell types do not occur in the planarian neoblasts. Understanding how the neoblasts keep such processes at bay could provide new insights into the etiology of aging, as well as extend our understanding of the processes involved in cellular maintenance.

One of the few proteins that are consistently and specifically present in the adult stem cells of such long-lived and highly regenerative animals are PIWI proteins (Sturm et al., 2017; van Wolfswinkel, 2014). These RNA-binding proteins use a small non-coding RNA called a piRNA to guide them to specific nucleotide sequences and suppress these, by inducing mRNA degradation or chromatin-based silencing (Czech et al., 2018; Haase et al., 2024). Their best-known targets are transposon sequences, but many other targets have been proposed. Whether these proteins have a role in upholding the negligible senescence of long-lived animal systems and, if so, how that would be accomplished, remains unknown.

We recently studied the role of the stem cell-specific PIWI protein SMEDWI-1 in the functioning of planarian neoblasts. We found that in the absence of SMEDWI-1, transposons remain repressed, but the RNA health of the neoblasts is no longer upheld (Allikka Parambil et al., 2024). Without SMEDWI-1, various non-coding and dysfunctional RNAs accumulate in the neoblasts without significant changes to the transcription of these molecules, indicating that SMEDWI-1 functions in the post-transcriptional quality control of these RNA transcripts.

Here, we report that SMEDWI-1-depleted animals over time develop a macroscopic phenotype that compromises their long-term health: upon extended knockdown of SMEDWI-1, animals acquire inefficiencies in wound closure, show ineffective regeneration, and become fragile, reminiscent of the deterioration of wound repair

and regeneration that commonly occurs in aging systems. Our analysis of this phenomenon points towards defects in protein secretion as the underlying cause. The connection between RNA fidelity and protein secretion runs through the signal recognition particle (SRP), which is a ribonucleoprotein (RNP) complex whose stoichiometry and assembly is affected by the overall RNA defect in *smedwi-1(RNAi)* animals. Our data indicate that the misregulation of RNA integrity (such as by the loss of SMEDWI-1) induces widespread effects that extend to deregulation of proteostasis and can result in progressive organismal fragility. This study emphasizes the importance of non-coding RNA in cellular function, and the importance of RNA control mechanisms for maintaining long-term cellular and organismal health.

## RESULTS

### Depletion of SMEDWI-1 results in defective regeneration

SMEDWI-1 is a stem cell-specific PIWI protein that is commonly used as a stem cell marker in *S. mediterranea* (Palakodeti et al., 2008;

Reddien et al., 2005). We recently reported that SMEDWI-1 enhances the resilience of the neoblast population (Allikka Parambil et al., 2024). However, we noticed an additional phenotype when knockdown of SMEDWI-1 was continued beyond the typical 3-week time span. Sexual animals that received more than 2 months of *smedwi-1* RNAi treatment demonstrated frequent epidermal blistering, wound closure defects, arrested regeneration and lysis of tissue fragments. To represent the spectrum of regeneration phenotypes of *smedwi-1(RNAi)* animals, we created a phenotype scale where 3-day blastemas with normal size and morphology were assigned a score of 0, while blastemas with signs of rupture were represented by a score of 1, and wounds that failed to close at all were given a score of 2 (Fig. 1A). After 2 months of *smedwi-1(RNAi)* treatment, nearly half of the animals failed to close their wounds after amputation, and after 3 months this increased to over 80% (Fig. 1B). Follow-up of such animals showed that tissue fragments with unclosed wounds did not complete regeneration and eventually died. Wound closure defects were primarily observed among fragments with a large wound

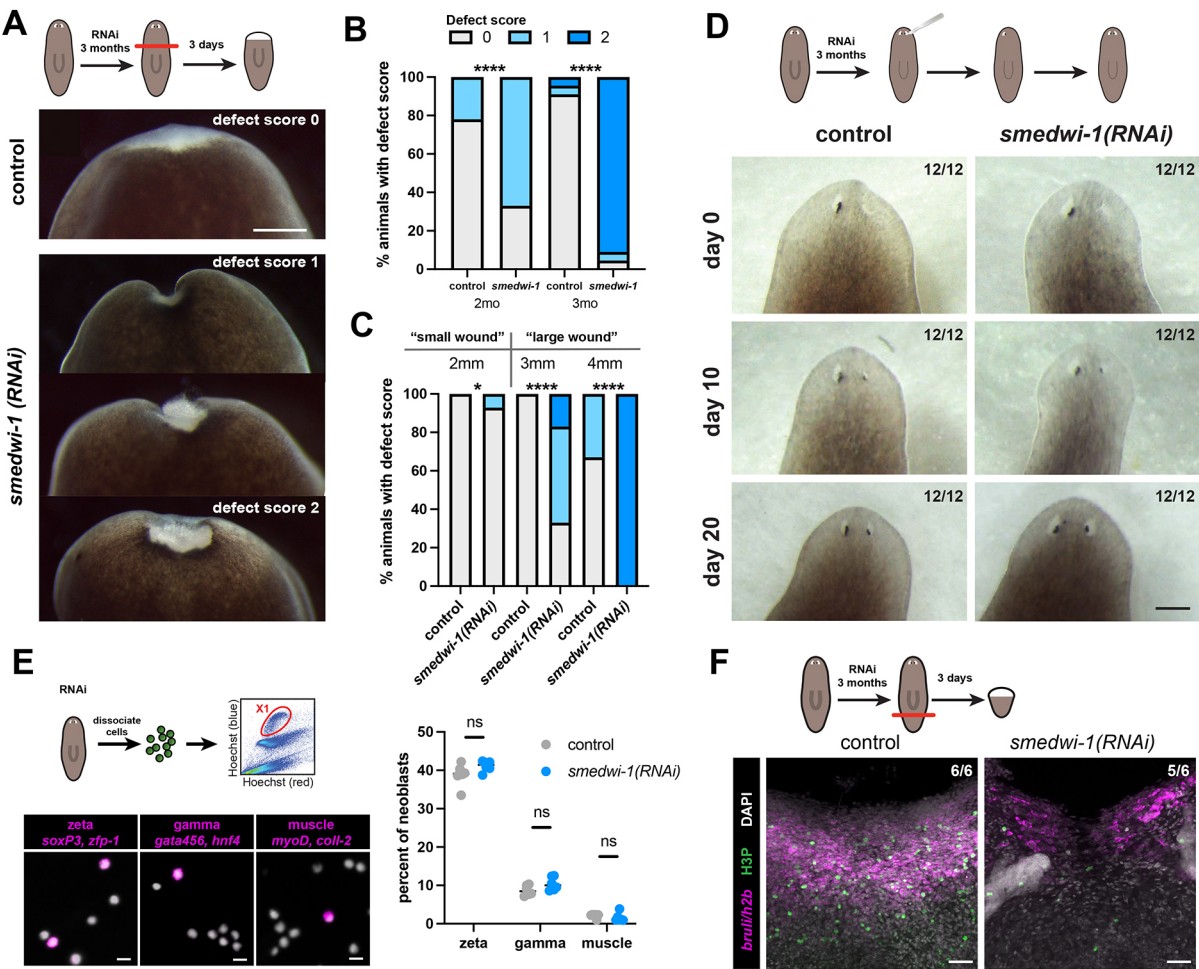

**Fig. 1. Loss of SMEDWI-1 leads to impaired wound healing.** (A) Live images showing the wound healing phenotypes at day 3 (d3) after amputation in long-term *smedwi-1(RNAi)* animals. Scale bar: 1 mm. (B) Quantification of observed wound-healing phenotypes at d3 post-amputation relative to the duration of the *smedwi-1(RNAi)* treatment (>50 animals per condition; Fisher's exact test, ****$P<0.0001$). (C) Quantification of observed wound-healing phenotypes at d3 post-amputation relative to the wound size in control and *smedwi-1(RNAi)* animals (>20 animals per condition; Fisher's exact test, *$P<0.05$, ****$P<0.0001$). (D) Live images showing the progression of eye regeneration in *smedwi-1(RNAi)* animals compared to controls. Scale bar: 1 mm. (E) Fluorescent *in situ* hybridization (FISH) on isolated neoblasts (X1) from *smedwi-1(RNAi)* animals and controls, showing several classes of lineage-specified neoblasts (left) and quantification (right). Scale bars: 10μm. Statistical analysis was carried out using a paired *t*-test; ns, not significant (6 biological replicates per RNAi, >100 cells analyzed per replicate). (F) Immunofluorescence of phosphorylated histone 3 (H3P) and FISH of *bruli* and *h2b* mark neoblasts at day 3 after amputation of large posterior fragments. *smedwi-1(RNAi)* fragments have reduced accumulation of neoblasts at the wound site. Scale bars: 50 μm.

diameter (Fig. 1C), and therefore were most apparent in large sexual animals. However, the regeneration defects were not due to effects on the germline, as *nanos(RNAi)* animals that lacked a germline (Wang et al., 2007) showed no defect in regeneration, whereas defects in *smedwi-1 nanos(double RNAi)* animals were comparable to those of size-matched *smedwi-1(RNAi)* animals (Fig. S1A). Further, phenotypes could be reproduced in asexual animals: the effect on transverse wounds was milder due to their smaller size (Fig. S1B), but parasagittal amputations confirmed a penetrant defect in *smedwi-1(RNAi)* asexual animals (Fig. S1C). The *smedwi-1(RNAi)* defect in wound closure and regeneration in asexual animals further presented as reduced ability of animal colonies to expand, as increased incidence of morphological defects, and as increased fragility in the face of microbial exposure (Fig. S2A-D). We therefore conclude that long-term depletion of SMEDWI-1 leads to defects in wound closure and blastema integrity, leading to organismal fragility.

### SMEDWI-1 depletion does not impair the proliferation or specification of neoblasts

We previously found that homeostatic expression of stem cell genes was not significantly altered in *smedwi-1(RNAi)* animals (Allikka Parambil et al., 2024). We therefore expected that the wound regeneration defect would not originate from general defects in stem cell function, but rather from the response to wounding. We made several observations that support this notion.

First, we confirmed that no major changes in neoblast transcripts or in transposon expression were detected upon long-term suppression of *smedwi-1* (Fig. S1D,E). Second, we found that minor wounds, such as incisions or eye ablations (LoCascio et al., 2017), were readily repaired in long-term *smedwi-1(RNAi)* animals (Fig. 1D), indicating that the neoblasts are still able to generate the cells that constitute the eye.

Third, we used fluorescent *in situ* hybridization (FISH) on isolated neoblasts to probe potential changes in the lineage biases of the neoblasts (van Wolfswinkel et al., 2014). Using markers of three major classes of neoblasts (epidermal, intestinal and muscle), we found that each of these classes could be readily detected and no significant changes in lineage contributions were found in the *smedwi-1(RNAi)* samples (Fig. 1E).

Fourth, we tested the ability of the neoblasts to respond to injury. Amputations that require tissue replacement trigger a wave of mitosis in the neoblasts around 48 h post-wounding, as well as migration of the neoblasts to the wound site (Wenemoser and Reddien, 2010). When evaluated in small tail amputations, *smedwi-1* animals were indistinguishable from control animals in their ability to mount this mitotic wave, as well as in the distribution of neoblasts through the tissue (Fig. S1F), indicating that neither the ability to cycle nor the ability to migrate was significantly affected. However, in large wound sites (>3 mm width), the density of mitotic cells, as well as the accumulation of neoblasts at the wound site were clearly reduced (Fig. 1F, Fig. S1G). Together, our data indicate that the regeneration defect in *smedwi-1(RNAi)* animals is not caused by loss of neoblast activity or lineage competency, but is related to specific properties of the wound.

### SMEDWI-1-depleted tissue fragments display a stalled wound response

The transcriptional response to wounding in *S. mediterranea* has been described in great detail and involves several waves of expression changes that primarily involve the epidermis, the muscle and the neoblasts (Wenemoser et al., 2012; Wurtzel et al., 2015). To determine whether the regeneration defect in *smedwi-1* animals

could be due to the inability to mount a wound response, we isolated wound sites at 6 h post-injury for RNA-sequencing analysis and determined expression of annotated wound response genes (Fig. 2A). The induction of many of these wound response genes was clearly detected, and no significant difference in gene expression was observed between the *smedwi-1* wound sites and the controls at this early time point. Over the days after amputation, the expression of wound response genes typically returns to baseline levels, as was indeed observed in control animals. However, in *smedwi-1* animals, the expression of several wound-response genes, including the early growth response genes *egr-2*, *egr-3* and *egr-like-1*, remained elevated for well over 1 week (Fig. 2B). The neoblast wound-response gene *runt-1* also remained elevated until at least day 10 after amputation, whereas general cell cycle-related neoblast genes did reduce their expression levels similar to the progression in wild-type neoblasts (Fig. 2C). Interestingly, positioning genes *wntP-1* and *wntless* also continued to increase in expression over time, instead of dropping back to baseline (Fig. 2D). Together, these data indicate that *smedwi-1(RNAi)* animals correctly initiate the early wound response and are able to activate the genes that are typical for the early phases of regeneration, but are unable to move beyond this stage.

### Connective structures are deregulated in *smedwi-1* animals

To elucidate why *smedwi-1* blastemas were unable to progress beyond the initial stages of wound regeneration, we inspected the tissue organization by applying cryosectioning on homeostatic samples and early wound sites. Control homeostatic samples showed a compact epidermal layer, that was tightly connected to a subepidermal layer of collagen (Fig. 2E, Fig. S2E-G). In *smedwi-1* animals, the epidermal layer was noticeably less tight, and more frequently dissociated from the underlying collagen. This phenomenon could already be recognized during fixation, as *smedwi-1* samples had a much higher likelihood of forming blisters or losing the epidermal layer altogether (not shown).

Upon amputation, control wound sites at 6 h post-injury showed early signs of repair, as has been previously reported (Morita and Best, 1974; Pedersen, 1976): while the wound edges remained clearly detectable, the wound surface was rapidly covered by a thin layer of collagen (Fig. 2F), and collagen filaments connecting the edges of the wound were readily observed (Fig. 2G,H). In *smedwi-1* wound sites, deposition of new collagen was still detected, but the collagen layer at the wound was thinner and more fragmented than in controls (Fig. 2F,H). The decrease in collagen in *smedwi-1(RNAi)* animals was confirmed by western blot analysis and levels were found to decrease over the course of several months (Fig. 2I), indicating that this effect builds up over time.

### *smedwi-1(RNAi)* animals misregulate transcripts of epidermal precursor cells

To determine the cause of these structural defects in *smedwi-1* wound sites, we analyzed whether loss of SMEDWI-1 resulted in any changes in the expression of tissue-specific genes (Fig. 3A). While most tissues appeared largely unaffected, we found a strong downregulation of genes specific to late epidermal precursors. We used the sub-classification of epidermal cells into 12 clusters, as established in a previous scRNAseq study (Fincher et al., 2018), to further dissect the cell-type specificity of this change in gene expression (Fig. 3B, Fig. S3A,B). Primarily, genes enriched in epidermal subclusters 4, 7 and 8 were reduced in *smedwi-1* samples. These three clusters are closely related and show high expression of genes that have previously been annotated as 'category 3 genes'

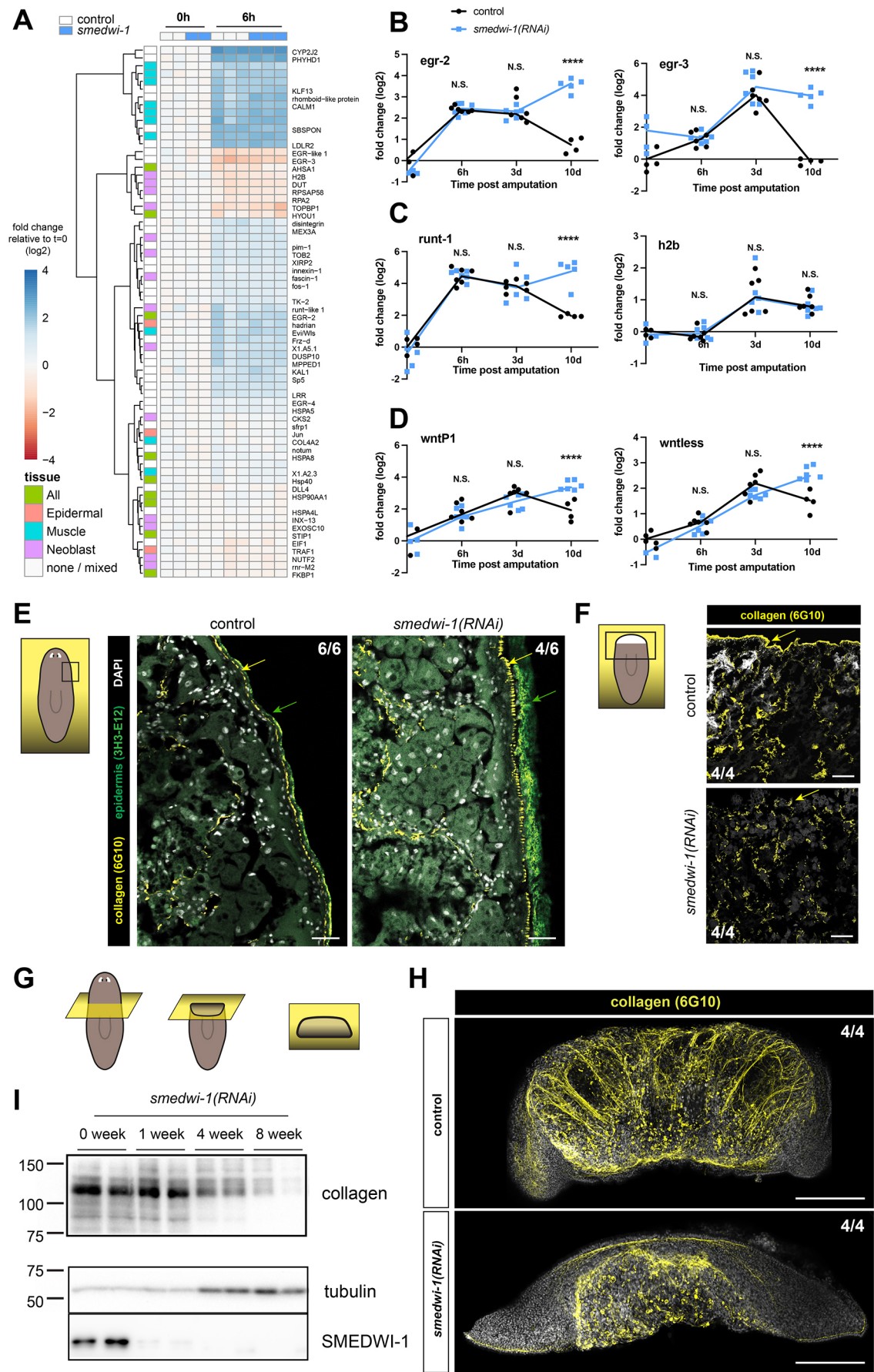

**Fig. 2.** See next page for legend.

**Fig. 2. Wound response is stalled and collagen is disorganized in**
***smedwi-1(RNAi)* animals.** (A) Heatmap showing the transcript levels
(log2 foldchanges relative to averaged 0 h controls as baseline) of previously
annotated wound-response genes in control animals and in *smedwi-1(RNAi)*
animals at 0 h and 6 h after wounding. For transcripts that are predominantly
expressed in one tissue (based on Wurtzel et al., 2015), the tissue is
indicated in the left column. Gene names based on best Blast hits to the
human proteome are provided unless no significant hit was found.
(B-D) qPCR analysis of a timecourse after amputation. Datapoints represent
individual biological replicates. Statistical analysis was carried out using an
unpaired *t*-test (****P<0.0001; N.S., not significant). (B) Two immediate early
genes show the expected increase in expression soon after wounding, but in
the absence of SMEDWI-1, the transcript levels remain elevated.
(C) Neoblast gene *runt-1*, which is involved in the early wound response,
shows the expected increase in expression soon after wounding, but in the
absence of SMEDWI-1, the transcript levels remain elevated. The general
neoblast transcript *h2b* is not affected by the absence of SMEDWI-1.
(D) Two positioning genes show the expected increase in expression soon
after wounding, but in the absence of SMEDWI-1, the transcript levels
remain elevated. (E) Immunostaining of coronal cryosections of homeostatic
*smedwi-1(RNAi)* and control animals shows disorganized collagen at the
basement membrane (yellow arrows) and disorganized epidermis (green
arrows). Scale bars: 50 µm. (F) Immunostaining of coronal cryosections of
*smedwi-1(RNAi)* and control 6 h wound sites shows strong reduction of
collagen covering the wound surface (indicated by yellow arrows). Scale bars:
100 µm. (G) Schematic illustrating the plane of view in H. (H) Immunostaining
of *smedwi-1(RNAi)* and control animals shows a mesh of collagen in control
animals, which is reduced and disorganized after long-term absence of
SMEDWI-1. Scale bars: 1 mm. (I) Western blots of homeostatic animals after
1, 4 or 8 weeks of *smedwi-1* knockdown shows the gradual loss of collagen
protein from the animals.

(cat3 genes) (Eisenhoffer et al., 2008). They are expressed in cells
that are located in the subepidermal layers that frequently have
protrusions towards the epidermis (Fig. 3C). Interestingly, markers
of early and mature epidermis, as identified in previous studies (van
Wolfswinkel et al., 2014; Wurtzel et al., 2017), were not
significantly affected (Fig. 3A), indicating that the effect likely
involves the lack of expression of the cat3 genes rather than a
complete block of the differentiation trajectory in the epidermal
lineage. Reduction of several cat3 genes was confirmed by qPCR
(Fig. 3D). Furthermore FISH analysis of intact *smedwi-1* animals
showed that the RNA level of category gene *agat-3* was indeed
reduced and, in particular, was lost from the boundary region,
whereas the mature epidermal gene *PRSS12* was not significantly
affected (Fig. 3E).

In addition to this stage-specific defect in the epidermal lineage,
we observed downregulation of some muscle and cathepsin cell
transcripts in the *smedwi-1* samples (Fig. 3A). FISH-based
detection of cells expressing the muscle-specific transcript
*collagen-2*, as well as immunostaining with the muscle-specific
TMUS antibody that labels myosin heavy chain (Cebria et al., 1996)
showed that muscle cell density was not reduced in the *smedwi-1*
samples (Fig. 3E), supporting the observation that only specific
transcripts were affected. However, sub-clustering failed to reveal
further cell-type specificity of these gene changes.

We wondered whether the reduction in cat3 epidermal transcripts
could be the cause of the *smedwi-1* wound repair defect. From
available single cell sequencing data, we found that the transcription
factor EGR-5 is specific to the cat3 cells (Fig. S3C), and in agreement
with a previous study (Tu et al., 2015), knockdown of EGR-5 resulted
in a reduction of transcripts for the cat3 genes, without affecting the
levels of transcripts that mark the mature epidermis (Fig. 3F).
Interestingly, knockdown of EGR-5 also resulted in a delay in wound
closure and failure of regeneration in 40% of large animals (Fig. 3G),
mimicking the effect of *smedwi-1* reduction on a shorter timescale.

This suggests that reduction of these cat3 transcripts could underlie
the *smedwi-1* wound healing phenotype.

## Loss of SMEDWI-1 results in destabilization of transcripts for secreted proteins

To determine whether the changes in the transcript levels were the
result of transcriptional or post-transcriptional regulation, we
performed qPCR for a subset of transcripts, using primers in the
mature mRNA as well as in introns, which are only present in the
nascent transcript (Fig. 3H). We found that, although levels of
mature mRNAs of epidermal genes were decreased, pre-mRNA
levels remained largely unaffected, indicating that the effect was on
the post-transcriptional level.

SMEDWI-1 is a member of the family of PIWI proteins. These
proteins bind small RNAs that can guide the PIWI to RNA targets by
sequence complementarity, leading to post-transcriptional regulation
of these targets. We thus considered whether the affected epidermal
transcripts would be direct targets of SMEDWI-1-bound small
RNAs. However, piRNA-mediated regulation typically leads to
gene silencing, which is opposite to the effects observed here.
Additionally, we found that epidermal sequences were depleted from
the SMEDWI-1-bound piRNAs (Fig. S3D). Together, this indicates
that the observed effect is unlikely to be a result of piRNA-mediated
regulation.

While inspecting the sequences of the destabilized transcripts, we
noticed that many of them encoded an N-terminal signal sequence
that labels proteins for secretion. We used computational analysis
to assign each protein a score for the likelihood of a signal
sequence (Hiller et al., 2004), and found that the affected sequences
were biased towards secreted proteins (Fig. 3I). Indeed, when we
separated the epidermal transcripts into those with and those without
a signal sequence, we found that specifically the transcripts of the
predicted secreted proteins were affected by loss of SMEDWI-1
(Fig. 3J). Similar results were obtained for the affected transcripts in
muscle cells and cathepsin cells (Fig. S3E,F). The production of
many of these transcripts (and presumably proteins) specifically in
the cat3 cells suggests that these cells function as secretory protein
factories in the epidermal lineage.

### *smedwi-1(RNAi)* animals show defects in epidermal secreted structures

The transcripts affected by loss of SMEDWI-1 largely encoded
small secreted proteins that did not resemble annotated protein
domains by BLAST, and appeared to be specific to planarians.
These features fit well with the proposed composition of planarian
rhabdoids. Rhabdoids are epidermal rod-shaped granules that
characterize the Rhabditophora to which planarians belong, and
may be secreted upon stress and injury to assist in physical barrier
formation and microbial defense (Hayes, 2017; Smith et al., 1982;
Wrona, 1986) (Fig. S4).

To examine the presence of rhabdoids in the epidermis
of *smedwi-1(RNAi)* animals, we examined transverse sections
stained with Hematoxylin and Eosin (H&E). This confirmed
several previously observed phenotypes, including a diminished
basement membrane in *smedwi-1(RNAi)* samples (Fig. 4A).
Further, the epidermal cells of the *smedwi-1(RNAi)* animals
appeared taller than those of control animals, and their structure
and contents appeared disorganized. These observations were
confirmed by electron microscopy images that, in addition to the
diminished basement membrane and disorganized cellular structure,
showed a reduction of rhabdoids in the *smedwi-1(RNAi)* epidermal
cells (Fig. 4B).

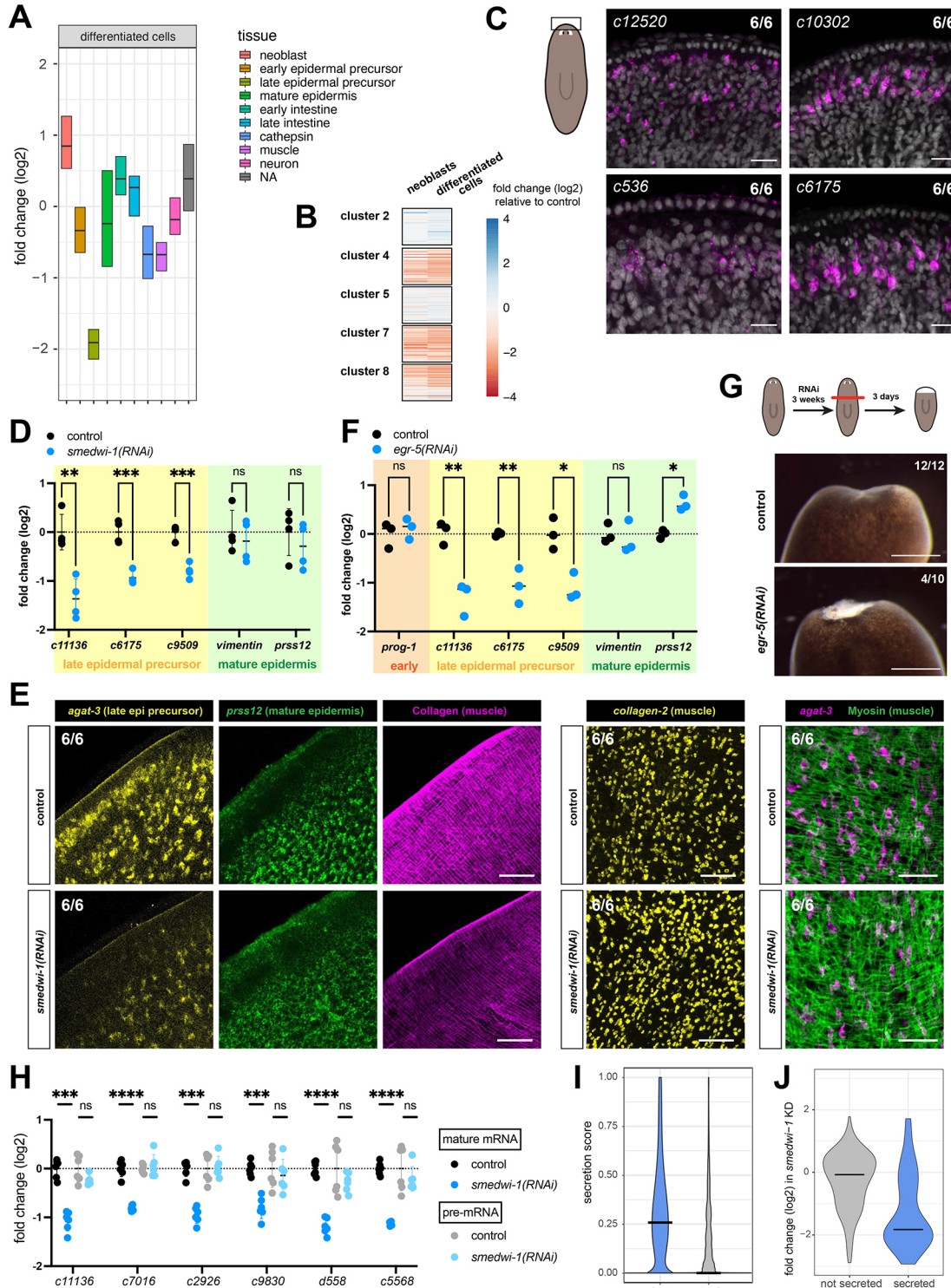

**Fig. 3.** See next page for legend.

We analyzed the rhabdoids secreted from the planarian epithelium upon salt stress as well as in the mucus trails of control and *smedwi-1(RNAi)* animals. The secretions of control animals contained rhabdoids of around 10 μm long (Fig. 4C), which rapidly inflated and formed a mucous network (Fig. S4A-C). Similar structures were identified in *smedwi-1(RNAi)* secretions, but these rhabdoids disintegrated much faster than control rhabdoids, leading to significantly reduced numbers of recognizable rhabdoids in their

mucus trails (Fig. 4C,D). We further found that the control rhabdoids were heavily covered in glycosylated molecules, as detected with the lectin Lens Culinaris Agglutinin (LCA), whereas the *smedwi-1(RNAi)* rhabdoids remained largely unstained (Fig. 4E, Fig. S4C), indicating that their molecular makeup was altered. Together, this indicates that various secreted protein structures were reduced or altered in the *smedwi-1(RNAi)* animals compared to controls.

**Fig. 3. Transcripts downregulated in the absence of SMEDWI-1 predominantly encode secreted proteins.** (A) RNAseq analysis of isolated differentiated cells from control and long-term *smedwi-1(RNAi)* animals shows reduced expression of transcripts that are characteristic of epidermal precursors, but not those specific to mature epidermis. Transcripts were assigned to cell types based on single cell RNAseq data (Fincher et al., 2018). (B) Analysis of subclusters of epidermal cells, as identified by Fincher et al. (2018), shows that transcripts reduced in *smedwi-1(RNAi)* animals are largely confined to the clusters that correspond to the late epidermal precursor stage, known as 'category 3 cells'. (C) FISH of several transcripts that are reduced in the *smedwi-1(RNAi)* animals shows the subepidermal localization of the expressing cells. Scale bars: 20 µm. (D) qPCR on independent samples confirms that transcripts of late epidermal precursors, marked by yellow shading, are significantly reduced in long-term *smedwi-1(RNAi)* animals. Mature epidermal transcripts, marked by green shading, remain largely unaltered. Datapoints represent biological replicates (*n*=4) each consisting of three animals. (E) FISH and immunostaining of *smedwi-1(RNAi)* and control animals shows the presence of muscle cells and mature epidermal cells, but the reduced presence of *agat-3* (in late epidermal precursor cells) and reduced collagen. Scale bars: 50 µm. (F) qPCR analysis of *egr-5(RNAi)* animals shows that transcripts of late epidermal precursors, marked by yellow shading, are significantly reduced. Mature epidermal transcripts, marked by green shading, as well as a characteristic transcript of early epidermal precursors marked by orange shading, remain largely unaltered. Datapoints represent biological replicates (*n*=3). (G) Live images showing the wound-healing phenotype observed at day 3 after amputation in *egr-5(RNAi)* animals. Scale bars: 1 mm. (H) Quantification of mature mRNA and unspliced pre-mRNA of several affected transcripts by qPCR shows that pre-mRNA levels are unaltered, indicating a post-transcriptional effect. Datapoints represent biological replicates (*n*=6).
(I) Violin plot showing the distribution of predicted signal sequence scores (as determined by the PrediSi package; Hiller et al., 2004) for transcripts reduced in *smedwi-1(RNAi)* animals compared to the whole proteome.
(J) Violin plot showing the distribution of fold changes in *smedwi-1(RNAi)* animals relative to controls, of epidermal transcripts encoding secreted and non-secreted proteins. In D, F and H, statistical analysis was carried out using an unpaired *t*-test (*$P<0.05$; **$P<0.01$; ***$P<0.001$; ****$P<0.0001$; ns, not significant).

### *smedwi-1(RNAi)* animals show defective assembly of the SRP

An important factor in protein secretion is the signal recognition particle (SRP), which is required to efficiently dock ribosomes translating secreted proteins onto the endoplasmic reticulum (ER). SRP consists of several proteins that are connected by the non-coding 7SL RNA (Fig. 5A, Fig. S5A). One side of the complex features SRP54, which binds the signal sequence of the nascent protein and accomplishes the docking of the ribosome on the ER. The other side of the complex consists of SRP9 and SRP14, and is required to arrest translational elongation until the connection to the ER is established. Defects in SRP-mediated relocation of the ribosome activate a quality control mechanism known as regulation of aberrant protein production (RAPP), which leads to degradation of both the protein and the mRNA (Karamyshev et al., 2014). A defect in SRP function therefore could explain the observed reduction in the transcript levels of genes encoding these secreted proteins.

We had previously found that loss of SMEDWI-1 results in the deregulation of many non-coding transcripts in the cells (Allikka Parambil et al., 2024). In agreement with these observations, we detected an increase in the level of the 7SL RNA in *smedwi-1(RNAi)* cells by qPCR (Fig. 5B) as well as by FISH (Fig. 5C, Fig. S5B,C). To determine whether malfunctioning of the SRP complex might underlie the *smedwi-1(RNAi)* phenotype, we knocked down SRP54 and SRP9. Loss of SRP54 resulted in rapid animal death. *srp-9(RNAi)* animals had a mild homeostatic phenotype that involved signs of increased adhesion to the substrate during

locomotion. Interestingly, amputation of the *srp-9(RNAi)* animals revealed a penetrant wound regeneration defect that resembled that of the *smedwi-1(RNAi)* animals (Fig. 5D), as well as a similar decrease of rhabdoids in the mucus trails (Fig. S4D,E). Furthermore, loss of SRP9 resulted in a reduction of the mature transcripts of secreted proteins without strong effects on the pre-mRNA levels, similar to the phenotype of *smedwi-1(RNAi)* animals (Fig. 5E).

To evaluate the integrity of the SRP complex in *smedwi-1(RNAi)* animals, we probed the levels of the 7SL RNA and the SRP54 protein (Fig. S5D,E) over the course of the RNAi treatment. Northern blotting confirmed that the 7SL RNA was slightly increased in *smedwi-1(RNAi)* samples (Fig. 5F). On the protein side of the complex, however, we detected a gradual decrease in the level of the SRP54 protein, indicating that in *smedwi-1(RNAi)* animals the stoichiometry of the SRP complex is progressively disrupted.

We next used gel filtration to estimate the size of the SRP complex in control and *smedwi-1(RNAi)* animals (Fig. 5G, Fig. S5F). We detected complexes containing the SRP54 protein around 200 kDa, both in the control animals and in the *smedwi-1(RNAi)* samples. This is in agreement with the expected size of the complex based on its components. SRP54 is the last protein to associate with the complex after other components have already been bound (Grosshans et al., 2001; Politz et al., 2000), and thus the complex containing SRP54 reflects the complete SRP in both conditions. In control samples, the 7SL RNA was found in high molecular weight fractions, as well as in the fractions around 200 kDa, indicating that part of the RNA is present in larger complexes, but a significant fraction is present as part of the complete SRP. In *smedwi-1(RNAi)* samples, however, the 7SL was concentrated in the high molecular weight fractions, suggesting that most of the 7SL RNA was sequestered in high molecular weight (HMW) complexes or aggregates rather than included as part of the mature SRP complex. This is in agreement with the reduced levels of the SRP54 protein that are found in these samples. Together, these data indicate that the SRP complex is progressively disrupted in the *smedwi-1(RNAi)* animals, and that this may underlie the observed alterations in protein secretions, the reduced levels of specific mRNAs, and the wound-healing defects.

### Loss of SMEDWI-1 leads to defects in proteostasis and cellular health

Defects in protein secretion could also lead to mislocalization of protein products and the accumulation of protein aggregates in the cells. We wondered whether such defects in proteostasis could be detected in the long-term *smedwi-1(RNAi)* cells.

We used the 6G10 antibody to quantify by western blot the amount of collagen protein that was retained inside cells. Whereas the total levels of collagen were progressively reduced in *smedwi-1(RNAi)* animals (Fig. 2H), we detected an increase in the amount of collagen protein in isolated dissociated cells in the *smedwi-1(RNAi)* samples (Fig. 6A), indicating that some of this protein aberrantly accumulates inside *smedwi-1(RNAi)* cells. Additionally, we detected an increase in ubiquitylated protein in *smedwi-1(RNAi)* samples, indicating that elevated levels of aberrant proteins labeled for degradation are present in these cells (Fig. S6A).

To determine whether the cells respond to this accumulation of retained protein, we used qPCR to determine the expression of several genes involved in retaining proteostasis (Fig. 6B). We found that *smedwi-1(RNAi)* animals expressed significantly increased levels of the cytoplasmic heat-shock protein HSPA8, which is involved in protein maturation and refolding, and of the gene ATG1,

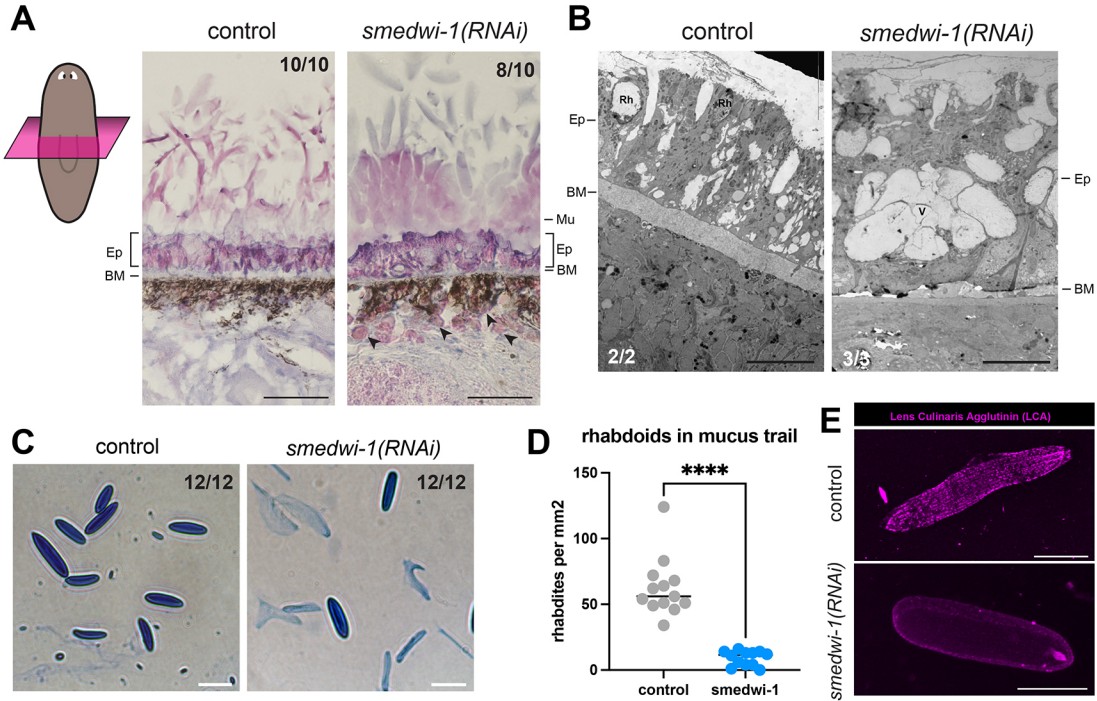

**Fig. 4. Secreted structures are affected in *smedwi-1(RNAi)* animals.** (A) Transverse sections stained with Hematoxylin and Eosin (H&E) show the altered structure of the epidermis in *smedwi-1(RNAi)* animals, including a less pronounced basement membrane (BM) and sub-epidermal accumulations of material (black arrowheads). Ep, epidermis; Mu, mucus. Scale bars: 50 µm. (B) Electron microscopy confirms the reduction of the basement membrane (BM) in *smedwi-1(RNAi)* animals and the absence of larger oblong rhabdoid structures (Rh). Instead, empty spaces (V) are found in the epidermis (Ep). Scale bars: 10 µm. (C) Anilin Blue staining of rhabdoids as secreted in the mucus trails of control and *smedwi-1(RNAi)* animals. (D) Quantification of the secreted rhabdoids indicates a reduction of rhabdoids in *smedwi-1(RNAi)* trails. Datapoints represent biological replicates (*n*=12-14). Statistical analysis was carried out using an unpaired *t*-test (****$P<0.0001$). (E) Staining of rhabdoids with the rhodamine-coupled lectin LCA shows that the composition of rhabdoids from *smedwi-1(RNAi)* animals is altered. Scale bars: 10 µm.

which is involved in autophagy of aggregates. Further, they also expressed elevated levels of the lysosomal gene β-galactosidase, which is typically upregulated in aged cells with defective proteostasis.

We next investigated whether protein aggregates could be detected in the *smedwi-1(RNAi)* cells. Previously, in the H&E stained samples, we had noticed an increase in large deposits of alkaline material in the subepidermal cells, which could indicate the accumulation of protein (Fig. 4A). We therefore applied the aggregate-specific protein dye Proteostat (Fig. S6B) to analyze the presence of protein aggregates in the epidermal tissues. We found that the *smedwi-1(RNAi)* epidermal cells stained more strongly with this dye than control epidermal cells (Fig. 6C, Fig. S6C). Further, the *smedwi-1(RNAi)* animals had more and larger accumulations of stained material in their subepidermal region, consistent with the presence of deregulated aggregates. Proteostat staining on isolated cells similarly showed aggregates inside the *smedwi-1(RNAi)* cells (Fig. 6D).

To more closely examine the subepidermal cells that were most strongly affected by the protein secretion defect, we used FISH staining for several cat3 transcripts to investigate the morphology of these cells (Fig. 6E). The subepidermal cells that express these transcripts remained present in the *smedwi-1(RNAi)* animals, but the transcript levels were reduced. Interestingly, the cells were also significantly larger and more irregular in shape (Fig. 6F), consistent with accumulation of protein material inside these cells. Similar effects were detected on isolated cells that stained with the cat3 probes (Fig. 6G).

Finally we used TUNEL staining on intact animals to quantify the incidence of cell death. We detected a significant increase in

apoptotic cells in the *smedwi-1(RNAi)* animals (Fig. 6H). Apoptotic cells were present throughout the animals, but were notably increased in the subepidermal region (Fig. 6I).

Together, our findings indicate that the deregulation of RNA quality control and alterations in non-coding RNAs, as caused by the absence of SMEDWI-1, can lead to far-reaching consequences that extend to loss of proteostasis, declining health of secretory cells and global defects such as reduced wound repair and accumulation of protein aggregates.

## DISCUSSION

Maintaining organismal health over time is a challenging task, as evidenced by the fact that most animals lose their vigor with progressing age, eventually resulting in death. While animals differ significantly in the length of their expected life, the latter part of their lifespan is typically plagued by a remarkably stereotypical set of phenotypes, such as impaired function of muscle and connective tissue, increased susceptibility to infection and impeded wound repair (Ashcroft et al., 2002; López-Otín et al., 2013, 2023). Changes in the efficiency of fundamental molecular processes likely underlie these degenerative phenomena, and indeed many aspects of the central dogma of gene expression are altered with age (López-Otín et al., 2013). Compared to their youthful counterparts, the chromatin of aged cells shows less distinction between accessible and silenced regions (Emerson and Lee, 2023; Yang et al., 2023). The transcription of repetitive regions that are kept suppressed in young cells becomes more pervasive in older cells (Ham et al., 2022; Pabis et al., 2024), leading to higher levels of transposon transcripts, and potentially to novel transposon insertions and genomic instability (Wood and

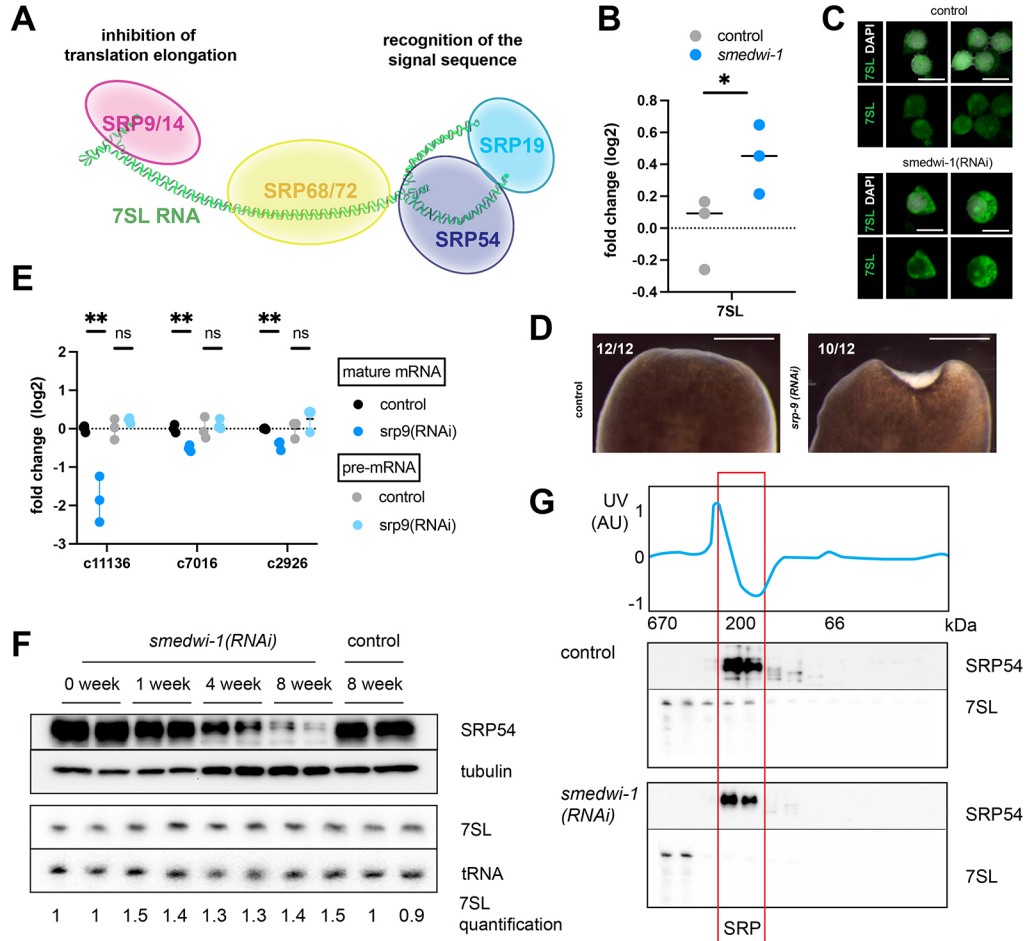

**Fig. 5. Absence of SMEDWI-1 leads to deregulation of the signal recognition particle.** (A) Schematic showing the composition of the signal recognition particle (SRP), indicating the 7SL RNA (green) as a scaffold that connects the SRP proteins. (B) qPCR data showing the increase in 7SL RNA in the absence of SMEDWI-1. Datapoints represent individual biological replicates ($n$=3). Statistical analysis was carried out using an unpaired $t$-test (*$P$<0.05). (C) RNA FISH for the *7SL* RNA on isolated cells shows increased intensity and prominent accumulation of the RNA in the cytoplasmic space in *smedwi-1(RNAi)* samples compared to controls. Scale bars: 10 μm. (D) Live images showing the wound-healing phenotype observed at day 3 after amputation of *srp-9(RNAi)* animals. Scale bars: 1 mm. (E) Quantification of levels of mature mRNA and unspliced pre-mRNA of several cat3 transcripts by qPCR shows that mature mRNA levels are reduced in the absence of SRP-9, whereas pre-mRNA levels remain unaltered, mimicking the effect of loss of SMEDWI-1. Datapoints represent biological replicates ($n$=3). Statistical analysis was carried out using an unpaired $t$-test (**$P$<0.01; ns, not significant). (F) Western blots and northern blots of animals after 1, 4 or 8 weeks of *smedwi-1* knockdown shows the mild increase of 7SL RNA, and the gradual loss of SRP54 protein from the animals, suggesting disruption of the SRP complex. Numbers below the northern blots indicate relative intensity of the 7SL band compared to the intensity of the tRNA band, as measured using the ImageQuant software. (G) Western blot and northern blot analysis of size fraction samples shows that only a minor fraction of 7SL RNA is found as part of the SRP complex in the *smedwi-1(RNAi)* animals.

Helfand, 2013). Further, proteostasis is altered between young cells and old cells: older cells tend to accumulate protein aggregates, which may impair the functioning of the cells (Huang et al., 2019; Taylor and Dillin, 2011; Walther et al., 2015). Whether these age-associated phenomena are interdependent, and which of them are causal to the downstream macroscopic phenotypes has been difficult to dissect, as the deregulation of these processes occurs concomitantly.

The planarian *Schmidtea mediterranea* normally has an indefinite lifespan and retains its ability to recover from any injuries. Yet here we find that, in the absence of the PIWI protein SMEDWI-1, planarians progressively lose their ability to regenerate and become increasingly fragile. This process takes place over multiple months after the elimination of the SMEDWI-1 protein is already complete, suggesting that this represents a progressive degenerative process, just like aging. Remarkably, the phenotypes that develop in these long-term *smedwi-1(RNAi)* animals also show resemblance to the phenotypes commonly observed in aging animals: we find defects in

wound healing, increased sensitivity to microbial exposure, alterations in protein secretion and ECM, accumulation of protein aggregates, and increased apoptosis – all of which are frequently found in aging systems.

We traced the progressive loss of regenerative ability in the *smedwi-1(RNAi)* animals to a defect in protein secretion. We found that loss of the RNA quality control mechanism spearheaded by SMEDWI-1 leads to uncontrolled accumulation of the 7SL RNA, causing misassembly of the SRP, which is essential for co-translational sorting of secreted proteins (Fig. 7A). Secreted proteins are essential for the formation of ECM, and play a particularly important role in epidermal protection and wound healing. The deregulation of protein secretion thus makes the animals less able to close wounds and regenerate, and more susceptible to environmental microbes. We find evidence that, in addition to a reduction in secreted protein, the dysfunction of the SRP also leads to accumulation of such proteins inside cells, which

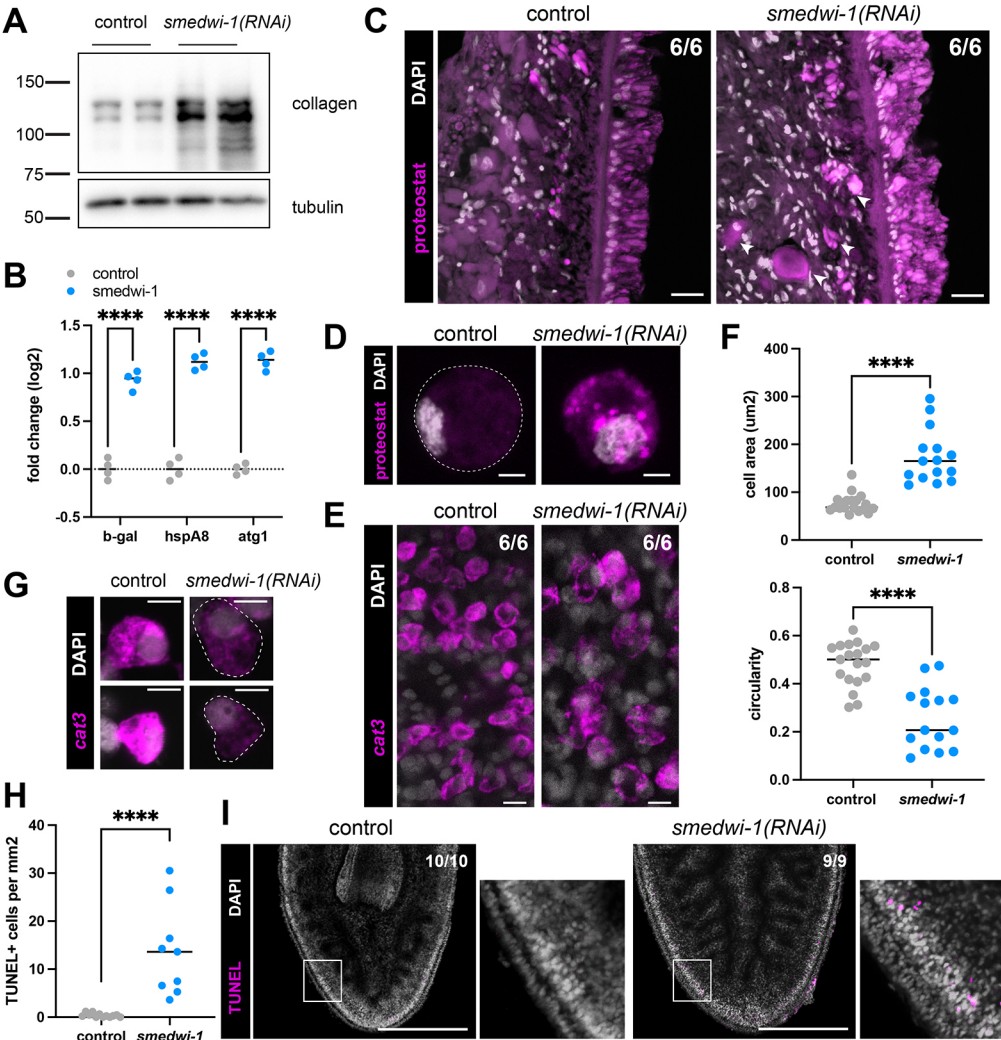

**Fig. 6. Proteostasis becomes deregulated in the absence of SMEDWI-1.** (A) Western blot of collagen on isolated live cells shows an increase of intracellular collagen in the *smedwi-1(RNAi)* samples. (B) qPCR data showing the increase in several proteostasis-related transcripts in *smedwi-1(RNAi)* samples. Datapoints represent biological replicates (*n*=4). Statistical analysis was carried out using an unpaired *t*-test (****$P$<0.0001). (C) Transverse sections stained with the aggregate-specific protein dye proteostat show the presence of protein accumulations in *smedwi-1(RNAi)* animals, in the epidermal cells and the subepidermal space (arrowheads). Scale bars: 20 µm. (D) Isolated cells stained with the aggregate-specific protein dye proteostat similarly show that *smedwi-1(RNAi)* cells contain aggregates within their cytoplasm. 2D projections of a *z*-stack through the entire cell are shown. Scale bars: 5 µm. (E) FISH for the cat3 transcript c6175 shows a lower transcript level and altered morphology of the expressing cells in *smedwi-1(RNAi)* samples compared to controls. Scale bars: 10 µm. (F) Quantification of the morphology of the cat3 cells based on the c6175 and c9509 FISH staining shows that, in *smedwi-1(RNAi)* samples, these cells are larger (top) and less regular in shape (bottom) than in controls. Datapoints represent individual cells (*n*=15-20). Statistical analysis was carried out using an unpaired *t*-test (****$P$<0.0001). (G) Isolated cells stained with RNA probes for cat3 genes based c6175 and c9509 similarly show that *smedwi-1(RNAi)* cat3 cells are more weakly stained, larger and less circular than cat3 cells from control samples. Scale bars: 10 µm. (H) Quantification of the density of apoptotic cells, as detected by TUNEL staining in homeostatic asexual animals shows a significant increase in cell death in *smedwi-1(RNAi)* compared to controls. Statistical analysis was carried out using an unpaired *t*-test (****$P$<0.0001). (I) TUNEL staining shows increased numbers of apoptotic bodies in *smedwi-1(RNAi)* animals. Apoptotic cells are primarily located in the subepidermal space (areas outlined are shown in the images to their right). Scale bars: 300 µm.

presents as aberrations in cell shape of secretory cells and formation of proteins aggregates (Fig. 7B). It is plausible that proteins that accumulate in the wrong cellular location are more likely to form aggregates; a recent study indeed found that secreted proteins and membrane proteins were major contributors to protein aggregates in aged cells (Chen et al., 2024).

While deregulation of chromatin and proteostasis have frequently been associated with aging cells and are part of the hallmarks of aging (López-Otín et al., 2013, 2023), RNA regulation is not typically evaluated in this context. To our knowledge, our findings form the first indication that defects in protein homeostasis with age

could actually (in part) derive from deregulation of RNA quality control, rather than from defects in protein-based mechanisms such as protein folding or autophagy. Notably, in several other systems, aging has been found to coincide with the accumulation of aberrant RNA (Adusumalli et al., 2019; Angelidis et al., 2019; Debès et al., 2023; Enge et al., 2017; Heintz et al., 2017; Kato et al., 2011; Kwon et al., 2023; Wang et al., 2020), indicating that this could be a widespread effect in aging systems.

Remarkably, a similar set of genes to the ones we found downregulated in *smedwi-1(RNAi)* animals has been reported to be downregulated upon the loss of the planarian cytoplasmic

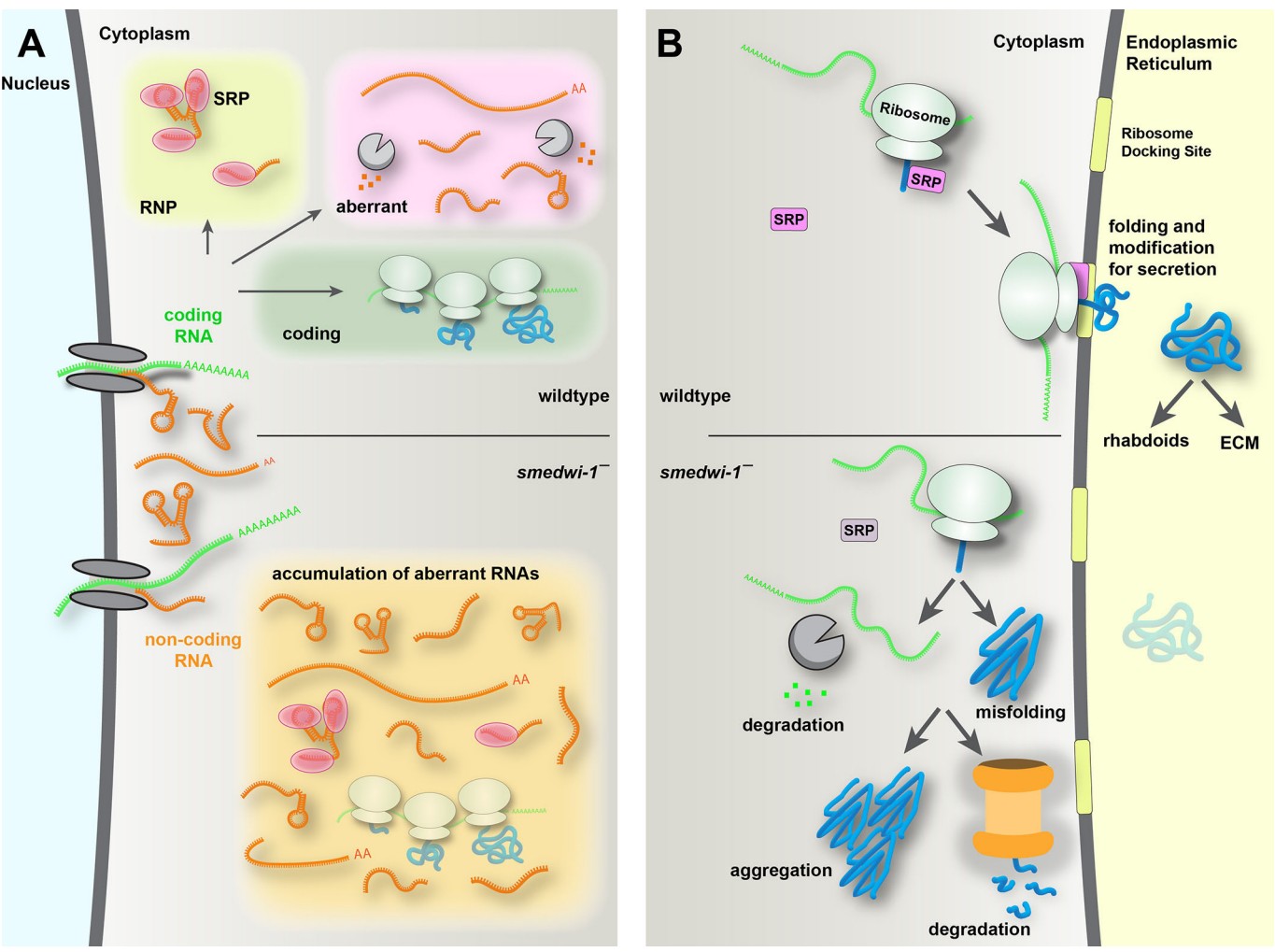

**Fig. 7. Model for the gradual loss of cellular health in the absence of SMEDWI-1.** (A) In wild-type stem cells, transcripts that are not coding or are mis-assembled with protein co-factors are routinely recognized and degraded. In the absence of SMEDWI-1, this selection process is defective, and aberrant RNAs can accumulate in the cells. Over time, this leads to a reduction in correctly assembled ribonuclear protein complexes (RNPs) such as the signal recognition particle (SRP). (B) The SRP functions to guide ribosomes that translate secreted proteins to the endoplasmic reticulum to continue protein translation and allow for appropriate folding and transport of the protein product. If such a ribosome is not correctly relocated, the protein product is released in the cytoplasm and marked for degradation together with the mRNA that produced it. Alternatively, the protein product may escape degradation and accumulate in the cytoplasm as a protein that is prone to misfolding and aggregation.

polyA-binding protein PABPC (Bansal et al., 2017). When comparing these two datasets, we found that specifically the transcripts encoding secreted proteins characteristic of the late epidermal precursor stage are reduced in the RNAseq data of both *smedwi-1(RNAi)* and *pabpc(RNAi)* samples (Fig. S6D). In addition, a regeneration phenotype and a prolonged wound response similar to those found in *smedwi-1(RNAi)* have been reported for *pabpc(RNAi)* animals, suggesting that these phenotypes may well be related. It is possible that PABPC is required for the stability of the mRNAs for these secreted proteins, or that it otherwise assists in the direction of these transcripts to the SRP and the ER-coupled translation. Alternatively, the loss of PABPC may result in broader anomalies in RNA control mechanisms that extend to non-coding RNAs and RNPs.

PIWI proteins are present in long-lived cells in a wide range of contexts. They are best known for their presence in the germline, but are also maintained in the long-lived stem cells of all studied highly regenerative animals (van Wolfswinkel, 2014), as well as in mesenchymal and hematopoietic stem cells in mammals (Sharma et al., 2001; Wu et al., 2010). Interestingly, PIWI proteins have

also been detected in neurons, which are some of the most long-lived post-mitotic cells (Nandi et al., 2016; Perrat et al., 2013; Rajasethupathy et al., 2012). It is possible that, in addition to their well-known role in transposon control, PIWI proteins function to maintain RNA health in each of these cell types and thereby mediate the longevity of these systems.

The PIWI protein SMEDWI-1 was one of the first genes identified as enriched in the planarian stem cells (Reddien et al., 2005). It has been widely used as a stem cell marker, but its role in planarian biology remained unclear as elimination of SMEDWI-1 did not result in a direct organismal phenotype. We now propose that it is exactly this lifelong presence of SMEDWI-1 in the neoblasts that bestows the neoblasts, and with them the planarians, their unusually long lifespan and regenerative abilities. SMEDWI-1 is not essential for the direct maintenance or replicative abilities of the stem cells, but it safeguards the RNA health of these cells and their descendants, thereby maintaining their lifelong functionality. Based on our findings we propose that, without SMEDWI-1, the planarians would become mere mortals, similar to

most other animals, and would probably experience similar age-related decline.

### Study limitations

This study focused on defects in the SRP in the *smedwi-1(RNAi)* animals, but it is possible that other RNPs are also affected by the deregulation of RNA control in the absence of SMEDWI-1. Misassembly of various RNPs may contribute to the decay in cellular health in these animals and may deregulate the intercellular communication to the stem cells, which could add to the misregulation of regeneration. Further, the mechanism by which loss of SMEDWI-1 causes the accumulation of aberrant RNA has not been resolved in this study.

The rod-shaped structures secreted from the planarian epidermis are a heterogenous set of granules. We were not able to conclusively determine that they would all fall under the term 'rhabdites' and thus chose to refer to them as rhabdite-like structures or 'rhabdoids'.

The exact epitope of the 6G10 antibody remains unknown. Based on our data, the most likely target is a collagen, and we have interpreted the labeling accordingly.

## MATERIALS AND METHODS

### Animal husbandry

*Schmidtea mediterranea* asexual clonal strain ClW4 and sexual strain S2 were maintained as previously described (Newmark and Sánchez Alvarado, 2000). Briefly, animals were cultured in 1× Montjuic salts at 20°C, fed homogenized beef liver paste every 1-2 weeks, and expanded through continuous cycles of amputation or fissioning and regeneration. Animals were starved 1-2 weeks prior to experiments. During RNAi experiments, animals were maintained in water supplemented with gentamycin to prevent bacterial growth. Experiments in this manuscript were performed on sexual animals, except where explicitly indicated that asexuals were used.

### RNAi

Regions of planarian genes 0.5-2 kb in length were amplified from complementary DNA (cDNA) using sequence-specific primers (Table S1) with adaptor sequences. The PCR product was cloned into the pGEM-T vector (Promega) and verified by Sanger sequencing. Both RNA strands were synthesized *in vitro* from PCR-generated forward and reverse templates with flanking T7 promoters (TAATACGACTCACTATAGG), and annealed by incubation at 37°C for 30 min. The transcribed ssRNA as well as the final dsRNA product were verified by gel electrophoresis.

Animals were starved 1-2 weeks prior to RNAi experiments. RNAi food was prepared by mixing 2 µl of generic food coloring, 2 µl of dsRNA and 50 µl of homogenized beef liver (Rouhana et al., 2013) and fed to animals in 3-day intervals for a total of five feeds unless noted otherwise. DsRNA matching *C. elegans* gene *unc-22* was used as a negative control.

### Whole-mount fluorescent *in situ* hybridization and immunofluorescence

Fixations, whole-mount *in situ* hybridization (ISH) and immunofluorescence were performed as previously described (Pearson et al., 2009), with alterations described by King and Newmark (2013). Briefly, formaldehyde fixed animals were bleached using formamide bleach solution and treated with proteinase K (2 µg/ml) in PBSTx. For FISH, following overnight hybridization at 56°C, samples were washed sequentially in pre-hyb solution, 1:1 pre-hyb-2×SSC, 2×SSC and 0.2×SSC at 56°C. Probes were detected with anti-DIG-POD (Roche, 11207733910), anti-Fl-POD (Roche, 11426346910) or anti-DNP-HRP (Perkin Elmer, PF1129). After tyramide development (King and Newmark, 2013), peroxidase was inactivated by incubation in 1% sodium azide. Specimens were counterstained with DAPI (Sigma). For immunofluorescence, animals were blocked and incubated with primary antibody overnight, followed by incubation with goat anti-rabbit IgG HRP conjugate (Life Technologies) or goat anti-mouse IgG HRP conjugate (Life Technologies). Primary antibodies used were rabbit anti-phospho-Histone3[Ser10] (Millipore, clone 63-1C-8; 1:750), mouse anti-collagen

6G10 (DHSB, Ross et al., 2015; 1:500), mouse epidermal antibody 3H3 (DHSB, Forsthoefel et al., 2014; 1:100) and mouse anti-myosin antibody TMUS (Cebria et al., 1996; 1:50). Signals were developed using Tyramide SuperBoost Kits (Invitrogen).

### Cyrosectioning and electron microscopy

For immunostaining, animals were fixed in 4% formaldehyde, followed by a dehydration series from 2% to 25% sucrose, and embedding in OCT (Tissue-Tek, Sakura). Sections were cut at 20 µm using a cryostat (Leica). Antibody incubations were performed as described above for whole mounts.

For H&E staining and electron microscopy (EM), animals were processed as described by Brubacher et al. (2014). Briefly, animals were fixed in fixative containing 2% formaldehyde, 2.5% glutaraldehyde, 54.2 mM cacodylate and 0.78 mM $CaCl_2$. Samples were then transferred to the EM facility or, for H&E staining, dehydrated in sucrose and embedded in OCT (Tissue-Tek, Sakura).

Proteostat staining was performed by incubation with a 1:1000 dilution of the dye in PBS for 10 min at room temperature followed by a 20 min destaining in 1% acetic acid (Shen et al., 2011). MG132 treatment (proteasome inhibitor for Proteostat control) was performed by soaking animals for 16 h in 1% DMSO in the presence of 100 µM MG132.

### Rhabdoid isolation and staining

For quantification of rhabdoids in mucus trails, five animals were left to roam in a 3 cm dish for 2 h. After removing the animals, the mucus left in the dish was stained in 5% Aniline Blue in PBS for 10 min, rinsed in PBS and imaged immediately for counting.

Images of rhabdoids were obtained by placing an individual animal on a microscope slide, removing remaining water and adding 20 µl 5 M NaCl. After 10 s, the animal was removed, and the secreted rhabdoids were stained by addition of 2 µl 5% Aniline Blue or 1% Lens Culinaris Agglutinin coupled to Rhodamine (Vector Systems) and covered with a coverslip. Samples were imaged immediately.

### Neoblast isolation and staining

Neoblasts in G2/M phase (X1) and differentiated cells (Xins) were isolated by fluorescence-activated cell sorting based on DNA content (Hoechst fluorescence), as reported by Hayashi et al. (2006), following procedures described previously (van Wolfswinkel et al., 2014).

For staining of the isolated cells, cell suspensions isolated by FACS were collected in CMFB and centrifuged at ~300 *g* for 5 min at 4°C. Cells were washed in CMF, spotted onto poly-D-lysine-coated coverslips (BD Biosciences), allowed to settle for ~30 min, and fixed in 4% PFA (in PBS) for 20 min at room temperature. Controls and treatment were always spotted on the same cover slip, and went through all staining steps in the same well. Immunofluorescence and FISH labeling were carried out similarly to the whole-mount protocol, with wash steps and antibody incubations shortened to 10 min and 1 h, respectively.

### Microscopy and image analysis

Images were taken on a Zeiss LSM800 confocal microscope. Control and RNAi animals were imaged with the same magnification, laser intensity and gain, at comparable anatomical position. Cell counting and quantification of fluorescence intensity were performed in Fiji (Schindelin et al., 2012). Quantification size and circularity of secretory cells was performed in PIQ (Allikka Parambil et al., 2024) using 2D images.

### qPCR analysis

Total RNA was isolated by Trizol and quantified by Qubit. To distinguish between changes in global RNA level or in level of polyadenylation, two cDNA preparations were synthesized from the same RNA samples: one primed by hexamers and one primed by oligo dT. For all cDNA preparations, ProtoScriptII (NEB) was used according to the manufacturer instructions, using 1 µg RNA as starting material in a 20 µl reaction. cDNA was diluted 1:5 in MilliQ water and 1 µl was applied to a 10 µl qPCR reaction using EvaGreen mastermix (Biotium). Primers are listed in Table S1. RT and qPCR reactions of samples and controls were run in parallel in the same plates. qPCRs were run on a QuantStudio 3 instrument

(ABI) with the following program: 95°C for 20 s; 40 cycles of 95°C for 5 s; 60°C for 20 s; followed by a melting curve analysis.

## RNA-seq library generation

For mRNA-seq libraries of neoblasts and differentiated cells, or regenerating tissue, large animals were fed control *unc-22* or *smedwi-1* dsRNA in liver for 2 months before FACS sorting or amputation. RNA was extracted from isolated tissues using TRIzol Reagent (Invitrogen), and libraries were generated using TruSeq RNA Library Prep Kit v2 (Illumina) following the manufacturer's instructions.

## Processing of mRNA-seq data

mRNA libraries were sequenced on a NovaSeq (Illumina). Additionally, previously generated data available under SRA accession number PRJNA905109 was included in the analyses. Reads were mapped against *Schmidtea mediterranea* transcriptomes WIX1 (van Wolfswinkel et al., 2014), dd_Smed_v6 (Liu et al., 2013) or unigene (Robb et al., 2015) using Bowtie2 (Langmead and Salzberg, 2012), and further processed with SAMtools (Li et al., 2009). For transposon expression analysis, reads were mapped against the *Schmidtea mediterranea* SMESG.1 genome (Grohme et al., 2018) using STAR (Dobin et al., 2013).

## Processing of small RNA-seq data

Small RNA libraries were reprocessed from (Allikka Parambil et al., 2024). Following adaptor trimming by Cutadapt (Martin, 2011), reads were mapped against the *Schmidtea mediterranea* SMESG.1 genome using Bowtie (Langmead, 2010), allowing for two mismatches and up to 20 mapping locations. The reads are then counted strand-sensitively toward exons of transcripts or transposon copies using BEDTools.

## Signal sequence prediction

The total *Schmidtea mediterranea* transcriptome was translated using ORFFinder (NCBI) to obtain a total proteome. The likelihood of a signal sequence in each protein was predicted using PrediSi (Hiller et al., 2004) (http://www.predisi.de/home.html), using default parameters. PrediSi calculates a normalized score on a scale between 0 and 1 for each protein sequence. A score greater than 0.5 means that the examined sequence likely contains a signal peptide. This was found in less than 5% of the planarian protein sequences.

## Size fractionation

Animals were rinsed in FPLC buffer (10 mM Tris (pH 7.5), 150 mM NaCl, 5 mM $MgCl_2$, 2 mM DTT, 0.2% NP-40 and 5% glycerol), and frozen in liquid nitrogen. After grinding and douncing, lysates were centrifuged at 20,000 *g* for 10 min. Supernatant was supplemented with RNase inhibitor and protease inhibitor cocktail (Complete, Roche), and 300 μl was loaded onto a Superdex 200 column (300/10, Cytiva) using a 1 ml loop on an AKTA Pure system. The column was calibrated with thyroglobulin (670 kDa), apoferritin (430 kDa), β-amylase (200 kDa), bovine serum albumin (66 kDa) and (to mark the total column volume) acetone (0 kDa). Fractions were collected after the void until the end of the column volume.

For analysis of protein content, each fraction was combined with sample buffer and loaded on a 12% protein gel. For analysis of RNA content, each fraction was supplemented with 1% SDS and an equal volume of acid PCI (phenol-chloroform-isoamylalcohol). The RNA in the aquatic phase was precipitated with 3 volumes of ethanol and 1 μl glycoblue.

## SDS-PAGE and western blotting

Individual 1-3 mm sized animals were homogenized in protein loading buffer (60 mM Tris-Cl at pH6.8, 5% glycerol, 1% SDS and 2.5% β-mercaptoethanol) and separated on 8% denaturing polyacrylamide gel. Samples were transferred to PVDF membrane, blocked and incubated with the primary antibody followed by secondary antibody, in PBSTw containing 1.5% milk. For the detection of ubiquitin, blots were denatured (6 M guanidinium chloride, 20 mM Tris at pH 7.5, 1 mM PMSF and 5 mM β-mercaptoethanol for 30 min) prior to blocking. The following antibodies were used: mouse anti-α-tubulin (MABT205, Millipore) at 1:10000; rabbit anti-SMEDWI-1 (Allikka Parambil et al., 2024) at 1:2000; mouse

anti-collagen (6G10, DHSB; Ross et al., 2015) at 1:1000-2000; rabbit anti-SRP54 (orb25807, Biorbyt) at 1:1000; mouse anti-myosin antibody TMUS (Cebria et al., 1996) at 1:500; rabbit anti-ubiquitin (P37, Cell Signaling Technology) at 1:300; goat anti-rabbit IgG HRP conjugate (Life Technologies) at 1:10,000; and goat anti-mouse IgG HRP conjugate (Life Technologies) at 1:10,000.

## Northern blotting

Samples were diluted with RNA loading dye (NEB) and separated on a 12% acrylamide gel. Gels were blotted to Hybond N+ membrane using a semidry apparatus (BioRad). Membranes were crosslinked at 12,000 $J/cm^2$ using a UV crosslinker (UVP), and incubated in ExpressHyb (Takara) containing end-labeled probe overnight at 42°C. After washes in 2×SSC with 0.05% SDS and 0.1×SSC with 0.1%SDS, membranes were exposed on a phosphoimager screen. Screens were read on a GE Typhoon FLA 9000 gel imager, and quantified using the imageQuant software.

Probes were ordered as DNA oligos and were end-labeled using gamma-ATP and PNK (NEB) for 1 h at 37°C. End-labeled probes were cleaned up using a G25 column (Zymo) before use for hybridization. Probe sequences are listed in Table S1.

## Quantification and statistical analysis

In all experiments shown, datapoints represent biological replicates. Mean values are indicated by a horizontal bar. Levels of significance were calculated with unpaired two-tailed Student's *t*-test, using the Prism software package unless otherwise indicated. Analysis of genome-wide data was carried out as described above.

## Acknowledgements

We thank the Keck DNA Sequencing Facility at Yale for Illumina sequencing, and the Center for Cellular and Molecular Imaging (CCMI) for electron microscopy of our samples. We further acknowledge the Yale Science Hill Imaging Facility and the Science Hill FACS Facility for providing access to confocal microscopes, cryostat and FACS equipment. We are grateful to Francesc Cebria for sharing the TMUS antibody. We thank members of the Van Wolfswinkel Lab for discussion and comments.

## Competing interests

The authors declare no competing or financial interests.

## Author contributions

Conceptualization: M.Z., D.L., J.C.v.W.; Formal analysis: M.Z., D.L., A.P., J.C.v.W.; Funding acquisition: J.C.v.W.; Investigation: M.Z., D.L., S.A.P., K.M., K.D., J.C.v.W.; Methodology: D.L., A.V., A.P.; Software: A.P.; Supervision: J.C.v.W.; Visualization: A.P.; Writing – original draft: J.C.v.W.; Writing – review & editing: M.Z., D.L., A.V., S.A.P., A.P.

## Funding

This work was supported by the National Institutes of Health (R35GM158281 and R01AG078926) and by the Vallee Foundation. Open Access funding provided by Yale University. Deposited in PMC for immediate release.

## Data and resource availability

Sequencing data generated in the course of this study have been deposited in the SRA under accession number PRJNA1209026. All other relevant data and details of resources can be found within the article and its supplementary information.

## Peer review history

The peer review history is available online at https://journals.biologists.com/dev/lookup/doi/10.1242/dev.204762.reviewer-comments.pdf

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
