## [Peer Review File · Development (Cambridge, England)]

Enhanced RNA quality control maintains long-term regenerative ability in planarians

Michael Zelko, Danyan Li, Andrew Verdesca, Sudheesh Allikka Parambil, Axel Poulet, Kaspar Mazeika, Krishnakali Dasgupta and Josien van Wolfswinkel
DOI: 10.1242/dev.204762

Editor: Mansi Srivastava

Review timeline

Original submission:	2 March 2025
Editorial decision:	1 May 2025
First revision received:	5 August 2025
Accepted:	4 September 2025

Original submission

First decision letter

MS ID#: dev.204762

MS TITLE: Enhanced RNA quality control maintains regenerative ability and prevents aging phenotypes in planarians

AUTHORS: Michael Zelko, Danyan Li, Andrew Verdesca, Sudheesh Allikka Parambil, Axel Poulet, Krishnakali Dasgupta and Josien van Wolfswinkel

Dear Dr van Wolfswinkel,

I have now received all the referees' reports on the above manuscript, and have reached a decision. The referees' comments are appended below, or you can access them online: please go to:

As you will see, the referees express considerable interest in your work, but have some significant criticisms and recommend a substantial revision of your manuscript before we can consider publication. If you are able to revise the manuscript along the lines suggested, which may involve further experiments, I will be happy receive a revised version of the manuscript. Your revised paper will be re-reviewed by one or more of the original referees, and acceptance of your manuscript will depend on your addressing satisfactorily the reviewers' major concerns. Please also note that Development will normally permit only one round of major revision. If it would be helpful, you are welcome to contact us to discuss your revision in greater detail. Please send us a point-by-point response indicating your plans for addressing the referees' comments, and we will look over this and provide further guidance.

Please attend to all of the reviewers' comments and ensure that you clearly highlight all changes made in the revised manuscript. Please avoid using 'Tracked changes' in Word files as these are lost in PDF conversion. I should be grateful if you would also provide a point-by-point response detailing how you have dealt with the points raised by the reviewers in the 'Response to Reviewers' box. If you do not agree with any of their criticisms or suggestions please explain clearly why this is so.

Reviewer 1

Advance summary and potential significance to field

Smedwi-1 is widely used as a stem cell marker in planarians. No physiological phenotype has been reported if smedwi-1 is reduced in expression. The authors found wound healing defects in planarians with prolonged smedwi-1 RNAi. They identified defects in epidermal cells, muscle, cathepsin and neoblasts. In addition, they found defects in protein secretion after smedwi-1 RNAi.

Comments for the author

1. The authors examined phenotypes of wound healing and regeneration in planarians with prolonged smedwi-1 RNAi. Data reported on wound healing has merit. The interpretation of the data in the framework of aging is unjustified, resulting in the conclusions unsupported by the data. The authors need to remove all texts related to aging in the title, abstract, introduction, and results, and interpretate the wound healing data as it is: what the phenotypes are when smedwi-1 is reduced.

Prolonged smedwi-1 RNAi phenotype does not equal to aging phenotype. One simple example in planarians is the prolonged RNAi of beta-catenin. Short-term RNAi leads to two heads. Long-term RNAi leads to multiple heads. We cannot interpretate the multiple head phenotype as aging phenotype. In the case of beta-catenin RNAi, it is an adaptation of stem cells to the defective patterning environments. In the case of smedwi-1 RNAi, it is likely a defect in muscle and epidermal which are critical for wound closure. These are not caused by aging.

2. Data from characterization of the phenotypic defects to potential downstream molecular effects after smedwi-1 RNAi contain at least four contradictory pieces of information. The authors need to explain why these images or data are contradictory to each other and improve the data presentation with reasonable justification of the corrections.

Fig. 3A: A significant number of genes were altered in neoblast after smedwi-1 RNAi. This is inconsistent with the author's statement that smedwi-1 RNAi did not impact neoblasts. In addition, stem cell proliferation and migration were reduced in Fig. 1F.

Fig. 4A is opposite to Fig. 2E. Both control and smedwi-1 RNAi showed thick layers of mucus in Fig. 4A, but not in Fig. 2E.

Fig. 6C is opposite to Fig. 3E. Fig. 6C shows increased collagen after smedwi-1 RNAi. Fig. 3E/ Fig. 2I shows reduced collagen after smedwi-1 RNAi. The authors' distinction between extracellular and intracellular collagen is not convincing. Collagen staining and WB in Fig. 3E/ Fig. 2I should also include intracellular.

Fig. 6A is opposite to Fig. 4A-D: The authors found that Proteostat can label rhabdoids strongly in both control and smedwi-1 RNAi samples. Fig. 6A shows higher rhabdoid content in smedwi-1 RNAi condition. Fig. 4A-D shows lower rhabdoid content in smedwi-1 RNAi condition.

3. The authors need to provide numbers of animals or cells or biological/technical replicates for all figures, including supplementary data.

4. Some images are of poor quality and low resolution. The authors need to use high quality images.

5. Statistic tests need to be provided for all data analysis.

6. Fig. 2E: The green arrow is more likely the mucus, instead of epidermis as stated by the authors.

7. "We recently reported that SMEDWI-1 enhances the resilience of the neoblast population (Allikka Parambil et al., 2024)."

Allikka Parambil et al. 2024 is a good study. However, there are no "stem cell resilience" experiments in this cited work.

8. "For some RNAi experiments, animals were maintained in water supplemented with Gentamycin to prevent bacterial growth."

Adding Gentamycin in some RNAi experiments can cause serious trouble in data interpretation. The authors need to provide specific notes in the results which figure or experiments used animals treated with gentamycin.

9. Fig. 2F: The collagen staining looks comparable between control and *smewi-1* RNAi, instead of what the authors stated, "disorganized collagen at the basement membrane". Due to large amounts of muscle in the planarians, collagen can be detected in many regions of the planarian histological sections. It is very hard to believe that the differences shown by the two images are true biological differences from *smewi-1* RNAi.

10. Fig. 2H: If wound healing is impacted, we should expect differences in collagen organization, which is an important component of planarian muscle. Hence, the difference shown by the images could be differences in muscle organization or contraction, consistent with defects in wound closure.

11. "When evaluated in small tail amputations, *smewi-1* animals were indistinguishable from control animals in their ability to mount this mitotic wave as well as in the distribution of neoblasts through the tissue (Suppl Figure S1E), indicating that neither the ability to cycle nor the ability to migrate is significantly affected. However, when amputated fragments with large wound sites were analyzed in a similar manner, the density of mitotic cells as well as the accumulation of neoblasts at the wound site were clearly reduced (Figure 1F, Suppl Figure S1F). Together, our data indicate that the regeneration defect in *smewi-1*(RNAi) animals is not caused by loss of the neoblasts' activity or lineage competency, but is related to specific properties of the wound." Images in Sup. 1E, 1F are of very poor quality, making the evaluation of these statements difficult. If the number of dividing neoblasts and migration of neoblast to large wound is reduced, this is a strong stem cell phenotype from *smewi-1* deficiency.

12. The authors need to clearly point out in their experiments whether sexual or asexual planarians were used.

13. "with a previous study (Tu et al., 2015) knockdown of EGR-5 resulted in a reduction of transcripts for the *cat3* genes, without affecting the levels of transcripts that mark the mature epidermis"

The authors' statement is inconsistent with the data. *prss12* is significantly different, which contrasts results from *smewi-1* RNAi.

14. "interestingly, knockdown of EGR-5 also resulted in a delay in wound closure and failure of regeneration in 40% of large animals (Figure 3G)," how are large animals and small animals defined?

15. No negative controls for qPCR experiments provided.

16. "we analyzed the rhabdoids in the mucus trails upon knockdown of the Signal Recognition Protein SRP9 and the late epidermal transcription factor EGR-5" Are rhabdoids co-localized with EGR-5 positive cells?

17. "most of the 7SL RNA is sequestered in large aggregates" is not supported by experimental data provided.

18. No orthogonal experiments to validate Proteostat results. The proteostat images appear difficult to rely on.

19. "We found that *smewi-1*(RNAi) animals express increased levels of the lysosomal gene *b-galactosidase*, which is a common marker of senescent cells" This could be due to *smewi-1*'s regulation of cathepsin cells which contain more lysosomal contents.

20. "Based on our data, deregulation of RNA control can directly cause global age-associated defects such as reduced wound repair and accumulation of protein aggregates.

This conclusion is unjustified. No direct data support this conclusion.

Reviewer 2

Advance summary and potential significance to field

In this study, the authors state that planarian worms knocked-down for SMEDWI-1 develop deficiencies in wound healing, which they interpret as a sign of ageing. Using these Smedwi-1(RNAi) animals as a model for studying aging, the authors observe that the extracellular collagen in the basal membrane is reduced, that the secreted epidermal structures are dysregulated and that the epidermal cells contain aggregates. At the molecular level, they identify a misregulation of transcripts in the late epidermis, namely a destabilization of the transcripts coding for secreted proteins, leading to a significant reduction in the number of rhabdoid epidermal structures. Finally, they propose that this secretion defect in Smedwi-1(RNAi) animals results from a misassembly of the Signal Recognition Particle (SRP) complex, with a progressive accumulation of 7SL RNA that becomes sequestered in large aggregates rather than being included in the mature SRP complex. The authors conclude that Smedwi-1 is a key actor in essential cellular mechanisms such as SRP and proteostasis, necessary to support cellular functioning and to prevent aging.

This study provides novel and significant results on the role of Smedwi in maintaining epidermal homeostasis in planarians by controlling protein secretion. The results as they currently exist are convincing and the figures are generally clearly presented.

Comments for the author

However, the authors' claims often go beyond the data they present:

- They show a convincing wound healing phenotype, but no data on whole body regeneration. Therefore, they should not write: "Depletion of SMEDWI-1 leads to defective regeneration".

- Similarly, in the absence of any characteristic sign of aging (decreased animal survival, decreased mitotic index of neoblasts, increased apoptosis, appearance of senescent cells, etc.), the link between the wound healing phenotype and the aging process has not been established in Smedwi-1(RNAi) animals. The authors cannot conclude their paper by writing: "In the absence of SMEDWI-1 the planarians become mere mortals, similar to most other animals, with similar age-related defects."

Therefore, either additional results must be added, or the presentation of the results and their discussion must be thoroughly revised before publication. Even in the absence of a reference to regeneration and aging, they will remain of great interest to cell biologists and developmental biologists.

A series of additional points need to be revised before publication:

- Regarding the efficiency of RNAi, what is the level of Smedwi-1 expression after 2 or 3 months of RNAi? Do the authors find a correlation between the intensity of the phenotype and the Smedwi-1 expression level?

- Why is the amputation level varying between experiments, i.e. anterior in Figure 1A and posterior in Figure 1F? Are the phenotypes identical?

- Failure to repair wounds may reveal an aging process as frequently observed in animals whose aging is well established, but it cannot be sufficient on its own to define an aging process. What is the organismal or cellular evidence that aging is actually at work in Smedwi-1(RNAi) animals?

- Is the gradual loss in collagen observed in homeostatic animals, independently of any amputation? This is not clear in Figure 2E, 2I.

- The heading (p9): "Smedwi-1(RNAi) animals misregulate transcripts of epidermal precursor cells" is confusing as in Figure 3A, transcripts specific to late epidermis are strongly down-regulated whereas transcripts from other cell types, including epidermal precursors, are not. Similarly, the legend of Figure 3A does not seem to reflect the results shown in this panel.

- In Figure 3E, the legend is either unclear or not appropriate as no epidermal precursor cells are shown

- In Figure 3D and 3F, it would be clearer to write directly on the figure the meaning of the color shadings

- In Figure 4A, more details on the histological view would be welcome; for example, what are the purple structures above the Basement Membrane?
- In Figure 4B, what are the large electron-light structures that are visible in the upper right panel? what does it mean "Rh?" in the lower right panel?
- The authors cite the extensive review on planarians as a model for understanding biological ageing (Sahu et al., 2017), but I would recommend that they also refer to the few experimental studies available on this question, e.g. Perrigue et al., 2015; Gambino et al., 2020; Deere et al., 2024
- This study describes cellular and molecular changes that take place exclusively in epidermal cells, therefore the authors should clarify in the discussion the two distinct roles played by Smedwi in planarians, in neoblasts on the one hand, and in the epidermal cells on the other hand.
- A summary scheme clarifying the link between the concomitant reduction in rhabdoid epidermal structures and the appearance of aggregates in late epidermal cells would be useful.
- How do the authors explain that Smedwi-1(RNAi) animals do not exhibit any organismal phenotype?

Reviewer 3

Advance summary and potential significance to field

In this paper, the authors have investigated more fully the phenotype of smedwi-1(RNAi) animals, a pleiotropic grouping of phenotypes that include altered RNA abundance that they originally uncovered in a prior paper (Allikka Parambil, et al). They investigate decreased wound healing in long-term smedwi-1(RNAi) animals, determining that stem cell abundance is unaffected, but that late epidermal progenitors are impacted. They also show that wound response genes are elevated long after amputation, suggesting failure to resolve the wound response. The authors also detect changes in cellular structures, including putative extracellular matrix and rhabdoids, before attempting to connect these ideas through the discovery that the non-coding RNA associated with the signal recognition particle is somewhat upregulated after smedwi-1(RNAi). Finally, the group uses Proteostat to test for possible protein aggregation in smedwi-1(RNAi) animals. Taken together, the group concludes that smedwi-1(RNAi) induces a suite of phenotypes overlapping with those seen in aging in other animals but not yet observed in planarians. Some of the ideas in this paper are really strong— understanding how Smedwi-1 regulates non-coding transcripts is very interesting and a potential understudied aspect of PIWI biology. Understanding why long-lived, regenerative species like planarians do not normally show signs of aging is also a fascinating research topic. However, enthusiasm for some of the goals of the research project is damped by a few considerable technical and interpretation issues. Thus, this paper does not yet meet the standards of rigor of the journal Development at this time. Major and minor concerns follow.

Major concerns:

- 1) The conclusion that the suite of phenotypes seen in smedwi-1(RNAi) animals represents "aging" is undersupported, particularly given some of the technical concerns with individual experiments (detailed below). One can imagine that a large number of perturbations in core RNA processing machinery might lead to a suite of outcomes that includes proteostasis issues, altered mRNA and protein production, and so on. At a minimum, the arguments laid out below would need to be strengthened to confirm that the phenotypes shown are a fair approximation of aging. Otherwise, the aging connection could be softened and moved to the discussion to mitigate this concern.
- 2) The 6G10 antibody was used to label "collagen" in Fig. 2 and throughout, but this antibody, developed by Ross and colleagues, is described in their paper as labeling muscles. It has also been used in the field by other researchers as a muscle marker. Therefore, the "collagen fibers" described in Fig. 2H are more likely to be "muscle fibers." It is also unclear what protein is being visualized in the western blots since to my knowledge the identity of the antigen that this antibody binds has not yet been determined. If the group has evidence that this antibody binds collagen (e.g. RNAi targeting a collagen gene reduces staining and western blot signal), that should be provided. Many of the conclusions drawn using this antibody should be revisited so that we can understand if it is the muscle that is affected or the ECM. If, indeed, the muscle is affected through the staining, then other results, like the differential expression of muscle genes might become more central to

the work. Alternatively, There is a collagen IV antibody used in Dubey, et al, which might be useful for testing ECM changes.

3) The primary impact of *smewi-1*(RNAi) occurring in epidermal and epidermal progenitor cells made me wonder whether *equinox* mRNA is impacted after *smewi-1*(RNAi). *Equinox* encodes a secreted protein from the epidermis and is absolutely required for blastema formation (Scimone, et al 2022 Nature Communications). Due to similarities in the long-term *smewi-1*(RNAi) phenotype and the *equinox*(RNAi) phenotype, it would be helpful to understand whether there is a direct connection between the two genes.

4) There are some inconsistencies between the size of "rhabdoids" seen in EM images (~ 2-3 μm long) and the stained mucus trail images (10 μm +) and in stained images (20-30 μm). Further, the Rh structures in Fig. 4B do not look a lot like rhabdites in other EM images (Hayes 2017 is a good reference). This left me wondering if the structures visualized are all "rhabdoids." Because these structures are not well characterized structurally or functionally in the literature, more evidence to support the consistent identity of these structures would help strengthen the argument that these structures are changed after *smewi-1*(RNAi). Are there other papers documenting that aniline blue or Lens Culinaris Agglutinin stain rhabdites/rhabdoids? Did you try Cupromeronic Blue (Hayes 2017)? Do they stain with microtubule antibodies as has been suggested in Lentz (1967) and Kornakova (2013)? Do rhabdoids have known function in regeneration or other "aging" phenotypes? For what it's worth, I do think they are really interesting and worthy of study!

5) The potential perturbation of 7SL in *smewi-1*(RNAi) was interesting, but there were a few issues with the size fractionation experiment. The UV graph at the top should have axes labeled. It is not clear how the group determined kDa assignments for the X axis when typically fractions are noted on this axis. It would be helpful to understand how the group assigned kDa values in this experiment given that flow-through rate of protein complexes can be impacted by both mass and shape of a complex. The UV graph also lacks some of the peaks seen in the Supplemental Figure so I am not sure how the graph in G was generated. More evidence would also be required to show that 7SL is present in aggregates rather than in higher-molecular weight, physiologically-relevant complexes.

6) Proteostat has, to my knowledge, not yet been validated in planarian cells. It would be helpful to see whether manipulations like MG132 treatment or chaperone knockdown result in increased proteostat staining, to confirm the validity of this tool in planarians. I was especially surprised to see a nuclear or perinuclear staining in Fig. A-B, given that most examples in the literature show Proteostat excluded from the nucleus. Another option here is to find a second strategy to support the conclusion that proteostasis is deregulated.

Minor concerns:

1) All antibodies that have not been validated in planarians before should be validated by RNAi and western blot experiments (e.g. SRP54) to verify specificity.

2) It is not always clear what statistical test was used for each experiment (e.g. Fig. 1E, right) and some statistical analyses may need to be reevaluated. N should also be provided for all experiments (e.g. Fig. 1F), either in the image or figure legend.

3) The white/DAPI staining of the *smewi-1*(RNAi) animal in Fig. 1F (and even clearer in the supplement) has some large white areas that don't look nuclear. Is this a real phenotype present in all animals in this experiment?? If so, what is this staining?

4) Can the authors please clarify why they called structures rhabdoids rather than rhabdites?

5) Does *smewi-1*(RNAi) cause eventual lethality? Given the emphasis on aging, this might be helpful to document.

Curiosity questions:

1) If the mature mRNA degradation in Fig. 3H is via RAPP, is it dependent on a planarian homolog of Ago2?

First revision

Author response to reviewers' comments

We thank the reviewers for their insightful comments. Based their feedback, we realized that the our use of the word “aging” in the title of this manuscript evoked strong reactions. We recognize that in a negligibly senescent system such as planarians there are no established phenotypes that mark aging, and we thus have altered the title of our manuscript to **“Enhanced RNA quality control maintains long-term regenerative ability in planarians”**.

We would like to emphasize that the ability to close wounds and recover from injuries are important features of adult organismal physiology. They reflect key developmental processes that are not commonly encountered during embryonic development, but are essential for maintaining organismal integrity over the course of post-embryonic life. A reduction in the ability to recover from injury leads to the loss of adult organismal integrity. Whether we call such a reduction in regenerative ability over time “aging” or just “progressive loss of organismal integrity” doesn’t change anything about the nature of the phenotypes, and thus doesn’t change the scope of the manuscript.

To further alleviate the reviewer concerns regarding the term “aging”, we have eliminated this term from the results section and the abstract, and reorganized the introduction to focus on maintenance of organismal health. We do maintain however that the suite of phenotypes that we find is remarkably reminiscent of aging phenotypes in other systems, and we would be amiss if we didn’t point this out.

The main point of our manuscript is that the rather subtle RNA control as mediated by the PIWI protein SMEDWI-1 is an important factor in maintaining post-embryonic health long-term in planarians. We find that loosening of RNA control can cause a significant subset of the phenotypes that are commonly found in aging systems, and that these phenotypes develop over extended time as would commonly be found for aging-related phenotypes. This is notable, as loss of RNA control is not generally recognized as a hallmark of aging, but nevertheless, based on our findings, could underlie a significant set of aging-associated phenotypes. Genes involved in RNA control, including PIWI proteins, are typically highly expressed during germ cell development and in early embryogenesis, but are lost after those stages. With our manuscript we aim to put out the hypothesis that the absence of this level of RNA control post-embryonically, may well be one of the causes for the gradual development of some of the phenotypes commonly associated with aging.

A point-by-point response is provided below.

Reviewer 1

SUMMARY OF THE ADVANCE MADE IN THIS PAPER AND ITS POTENTIAL SIGNIFICANCE TO THE FIELD

Smedwi-1 is widely used as a stem cell marker in planarians. No physiological phenotype has been reported if smedwi-1 is reduced in expression. The authors found wound healing defects in planarians with prolonged smedwi-1 RNAi. They identified defects in epidermal cells, muscle, cathepsin and neoblasts. In addition, they found defects in protein secretion after smedwi-1 RNAi.

SUGGESTIONS TO AUTHORS

1. The authors examined phenotypes of wound healing and regeneration in planarians with prolonged smedwi-1 RNAi. Data reported on wound healing has merit. The interpretation of the data in the framework of aging is unjustified, resulting in the conclusions unsupported by the data. The authors need to remove all texts related to aging in the title, abstract, introduction, and results, and interpretate the wound healing data as it is: what the phenotypes are when smedwi-1 is reduced.

Prolonged smedwi-1 RNAi phenotype does not equal to aging phenotype. One simple example in planarians is the prolonged RNAi of beta-catenin. Short-term RNAi leads to two heads. Long-term RNAi leads to multiple heads. We cannot interpretate the multiple head phenotype as aging phenotype. In the case of beta-catenin RNAi, it is an adaptation of stem cells to the defective patterning environments. In the case of smedwi-1 RNAi, it is likely a defect in muscle and epidermal which are critical for wound closure. These are not caused by aging.

We thank the reviewer for this perspective, and for their appreciation of our data on the effect of long-term *smedwi-1* knockdown on wound healing and regeneration. Wound healing and regeneration are post-embryonic processes required to maintain organismal health. As such, these processes are essential for extending adult lifespan, and the reduction of these processes inevitably results in the accumulation of time/age-related damages and a decline in organismal health over time/age.

Defects in wound healing and proteostasis that develop over time are commonly observed in other aging model systems, and for that reason we referred to them as “aging-related” or “aging-like” phenotypes. We have now clarified this in the manuscript. Indeed, such phenotypes are likely caused by defects in muscle, ECM, and/or epidermis that increase over time. This is exactly what we observe in the absence of SMEDWI-1.

2. Data from characterization of the phenotypic defects to potential downstream molecular effects after *smedwi-1* RNAi contain at least four contradictory pieces of information. The authors need to explain why these images or data are contradictory to each other and improve the data presentation with reasonable justification of the corrections.

We appreciate the opportunity to clarify these data as laid out below.

Fig. 3A: A significant number of genes were altered in neoblast after *smedwi-1* RNAi. This is inconsistent with the author's statement that *smedwi-1* RNAi did not impact neoblasts. In addition, stem cell proliferation and migration were reduced in Fig. 1F.

As indicated in Figure 3A, the gene expression changes shown are in differentiated cells, and not in neoblasts. The change in proliferation and migration detected in Figure 1F is unlikely to be caused by a defect in the neoblasts, but rather by a defect in signalling. The evidence for this is that 1. there are no major changes in gene expression in the homeostatic neoblasts (Suppl Figure S1C); 2. that lineage selection of *smedwi-1*(RNAi) neoblasts is similar to that of control neoblasts (Fig 1E); and 3. that the neoblasts proliferate and migrate normally in a different (smaller wound) context (Suppl Fig S1E).

Fig. 4A is opposite to Fig. 2E. Both control and *smedwi-1* RNAi showed thick layers of mucus in Fig. 4A, but not in Fig. 2E.

Mucus is easily separated from the animals during fixation. In fact, the first step in the most widely used planarian fixation protocol (King 2013) is to treat the animals with the mucolytic agent N-acetyl Cysteine (NAC). This has accomplished the removal of mucus in the images in Figure 2E. In the case of Figure 4A a different fixation protocol was used to accommodate the maintenance of structural integrity and the H&E staining (Brubacher 2014). This protocol does not use NAC, and thus the mucus was not removed. We would like to note however that we do not draw conclusions from the presence or absence of mucus in these images as, as stated above, mucus is easily separated from the animals, and quantification of the mucus thus would require different, dedicated methods.

Fig. 6C is opposite to Fig. 3E. Fig. 6C shows increased collagen after *smedwi-1* RNAi. Fig. 3E/ Fig. 2I shows reduced collagen after *smedwi-1* RNAi. The authors' distinction between extracellular and intracellular collagen is not convincing. Collagen staining and WB in Fig. 3E/ Fig. 2I should also include intracellular.

To accommodate the flow of the story, the Western blot of intracellular collagen is shown in Fig 6A (previously 6C) instead of adding this to Figure 3E. We agree that the finding of an increase of intracellular collagen while total collagen is reduced is surprising, but it is not contradictory. It merely reflects the fact that there is vastly more extracellular collagen than intracellular collagen.

Fig. 6A is opposite to Fig. 4A-D: The authors found that Proteostat can label rhabdoids strongly in both control and *smedwi-1* RNAi samples. Fig. 6A shows higher rhabdoid content in *smedwi-1* RNAi condition. Fig. 4A-D shows lower rhabdoid content in *smedwi-1* RNAi condition.

We appreciate the opportunity to clear up this misunderstanding.

1. We did not state that rhabdoids are lost in *smedwi-1*. We merely report that by EM the subepidermal structures that resemble rhabdoids are disrupted, and that secreted rhabdoids from *smedwi-1* animals disintegrate faster and contain less of the epitope (presumably glycosylated protein) that is recognized by the lectin LCA.

2. We did not find that Proteostat labels rhabdoids. We found that certain subepidermal structures are stained with the protein aggregation dye Proteostat, even in control animals. The localization and the elongated shape of a minority of these structures suggests that those might be developing rhabdoids. The majority of the stained structures in the *smedwi-1* animals however are larger and spherical protein accumulations, and additionally increased dispersed signal is detected in the mature epidermis. This is unlikely to reflect rhabdoids and rather reflects aggregated protein, as this is what Proteostat is known to label. To avoid confusion we have removed the phrase hypothesizing that the subepidermal structures in control samples might be rhabdoids, as we do not have actual evidence to support this.

3. The authors need to provide numbers of animals or cells or biological/technical replicates for all figures, including supplementary data.

We have now included this information. In general, we do not use technical replicates - all replicates are biological.

4. Some images are of poor quality and low resolution. The authors need to use high quality images.

This may have been an artifact caused by the pdf assembly process. We have now provided individual files of the figures to avoid this problem.

5. Statistic tests need to be provided for all data analysis.

We have now included specific labelling for each of the tests.

6. Fig. 2E: The green arrow is more likely the mucus, instead of epidermis as stated by the authors.

As explained above, these samples were treated with the mucolytic agent NAC as part of the fixation protocol and thus have likely lost their mucus layer. Additionally the antibody used for this labelling (3H3; Forsthoefel 2014) was described to label epidermis, and labels epidermis in our hands. We are not aware of any evidence that this antibody would label mucus.

7. "We recently reported that SMEDWI-1 enhances the resilience of the neoblast population (Allikka Parambil et al., 2024)."

Allikka Parambil et al. 2024 is a good study. However, there are no "stem cell resilience" experiments in this cited work.

We thank the reviewer for their appreciation of our previous work. For an illustration of the reduced stem cell resilience in the absence of *smedwi-1*, please refer to Figure 1E-F in that study, showing the diminished recovery of *smedwi-1(RNAi)* neoblasts from irradiation.

8. "For some RNAi experiments, animals were maintained in water supplemented with Gentamycin to prevent bacterial growth."

Adding Gentamycin in some RNAi experiments can cause serious trouble in data interpretation. The authors need to provide specific notes in the results which figure or experiments used animals treated with gentamycin.

This is an interesting perspective. We appreciate that the addition of an antibiotic can modify the microbiome of the planarians and may affect phenotypes related to microbial interactions. In our experience the removal of Gentamycin from the *smedwi-1* animals makes them more fragile and exaggerates some of the phenotypes that we observe, but does not create any

additional or substantially different phenotypes. The fragility of the *smewi-1* animals is such that without Gentamycin the long-term knockdown experiments would not be possible as the majority of the animals would be lost or disorganized. We have now added documentation of this effect in Supplemental Figure S2 and we have clarified in the methods that all experiments are conducted in the presence of Gentamycin unless indicated otherwise.

9. Fig. 2F: The collagen staining looks comparable between control and *smewi-1* RNAi, instead of what the authors stated, "disorganized collagen at the basement membrane". Due to large amounts of muscle in the planarians, collagen can be detected in many regions of the planarian histological sections. It is very hard to believe that the differences shown by the two images are true biological differences from *smewi-1* RNAi.

We agree that the collagen in the parenchyma is difficult to quantify, and in accordance we draw no conclusions regarding that. However, it is very clear that the control sample has a layer of collagen covering the wound surface, whereas in the *smewi-1*(RNAi) sample this layer is strongly reduced (see the arrows in Figure 2F).

10. Fig. 2H: If wound healing is impacted, we should expect differences in collagen organization, which is an important component of planarian muscle. Hence, the difference shown by the images could be differences in muscle organization or contraction, consistent with defects in wound closure.

Indeed, a defect in wound closure is detected in *smewi-1*(RNAi) animals, and we indeed detect a defect in collagen organization. We did not detect a significant reduction in the homeostatic density of muscle cells as detected by *collagen-2* mRNA, but we did detect reduction in collagen protein levels as detected by the 6G10 antibody. The image shown in Figure 2H is of 6h wound sites, and thus is among the earliest displays of the wound healing defect. We indeed interpret this as an inability of the muscle to cross and close the wound. We have clarified this in the legend.

11. "When evaluated in small tail amputations, *smewi-1* animals were indistinguishable from control animals in their ability to mount this mitotic wave as well as in the distribution of neoblasts through the tissue (Suppl Figure S1E), indicating that neither the ability to cycle nor the ability to migrate is significantly affected. However, when amputated fragments with large wound sites were analyzed in a similar manner, the density of mitotic cells as well as the accumulation of neoblasts at the wound site were clearly reduced (Figure 1F, Suppl Figure S1F). Together, our data indicate that the regeneration defect in *smewi-1*(RNAi) animals is not caused by loss of the neoblasts' activity or lineage competency, but is related to specific properties of the wound." Images in Sup. 1E, 1F are of very poor quality, making the evaluation of these statements difficult. If the number of dividing neoblasts and migration of neoblast to large wound is reduced, this is a strong stem cell phenotype from *smewi-1* deficiency.

As explained above, the reduction in neoblast accumulation at the affected wound sites is unlikely to reflect a defect in the neoblasts, as the neoblasts are able to migrate and divide in response to smaller wounds. The most likely explanation for the altered neoblast response thus is that the observed defect reflects a defect in signalling that originates from the wound site which is disrupted due to the inability to close the wound.

12. The authors need to clearly point out in their experiments whether sexual or asexual planarians were used.

We thank the reviewer for pointing out that this was not sufficiently clear, and have now clarified this in the Methods section. The entire study is performed on sexual animals with the exception of experiments that are explicitly indicated to have used asexuals.

13. "with a previous study (Tu et al., 2015) knockdown of EGR-5 resulted in a reduction of transcripts for the *cat3* genes, without affecting the levels of transcripts that mark the mature epidermis"

The authors' statement is inconsistent with the data. *prss12* is significantly different,

which contrasts results from *smewi-1* RNAi.

The reviewer is correct that the level of *prss12* differs between control animals and *egr-5(RNAi)* animals. The critical distinction made however is whether *egr-5* causes a block in epidermal differentiation or rather an alteration in one of the intermediate epidermal cell states without blocking further differentiation. While the levels of “cat3” genes are significantly reduced, the level of the mature epidermal marker *vim* is unaltered and the mature epidermal marker *prss12* is slightly increased in *egr-5(RNAi)* samples, which argues against a block in epidermal differentiation. Indeed, in the *smewi-1(RNAi)* samples neither of the mature epidermal markers changes. We do not currently know whether this reflects a substantial difference between the phenotype of *egr-5* and *smewi-1*, but it is well possible that the transcription factor EGR-5 regulates genes apart from the “cat3” genes encoding secreted proteins that are deregulated in *smewi-1(RNAi)*, and thus that knockdown of *egr-5* affects processes that extend beyond the effect of *smewi-1(RNAi)*.

14. "interestingly, knockdown of EGR-5 also resulted in a delay in wound closure and failure of regeneration in 40% of large animals (Figure 3G)," how are large animals and small animals defined?

We thank the reviewer for requesting this clarification. Animals that have a width of more than 3mm are considered large. We have clarified this in Figure 1C and in the text.

15. No negative controls for qPCR experiments provided.

We are not sure what the reviewer means by this comment. Our qPCR primer sets are routinely validated for quantitative amplification and for absence of signal on genomic DNA. No-template controls and no-RT controls do not give any signal and thus cannot be quantified. At best we could state that their level is below detection limit. It is very uncommon to add this to graphs.

16. "we analyzed the rhabdoids in the mucus trails upon knockdown of the Signal Recognition Protein SRP9 and the late epidermal transcription factor EGR-5" Are rhabdoids co-localized with EGR-5 positive cells?

We do not have an antibody to detect EGR-5, and thus do not know which cells contain the EGR-5 protein. *egr-5* mRNA is primarily present in late epidermal precursor cells (“cat3 cells”, as named after the staging in Eisenhoffer 2008). Rhabdoids are present in mature epidermal cells and some subepidermal cells. These cells do not overlap with cells expressing peak levels of *egr-5* mRNA although it cannot be excluded that low levels of *egr-5* transcript and/or EGR-5 protein are present in these cells.

17. "most of the 7SL RNA is sequestered in large aggregates" is not supported by experimental data provided.

Please consider Figure 5G. The 7SL RNA is present in the fractions right after the void, which represent the aggregates and very high molecular weight complexes. Maybe the reviewer meant that it is not possible to distinguish between these two organizations based on the size fractionation, and we did not intend to make this distinction. To remedy this, we have replaced “aggregates” by “aggregates or high molecular weight complexes”.

18. No orthogonal experiments to validate Proteostat results. The proteostat images appear difficult to rely on.

We indeed prefer to use multiple orthogonal methods to validate our findings. Unfortunately no methods were available to analyze protein aggregation in planarians, and in non-standard model systems it is unfortunately not always feasible to develop multiple new methods per data point. The detection of increased levels of collagen inside the cells confirms the intracellular accumulation of secreted protein, which is likely to form aggregates, and this can thus be considered an imperfect, but still orthogonal method. We now also added data showing the increase in ubiquitination of protein in the *smewi-1(RNAi)* animals (Supplementary Figure S6A)

as an additional indication that misfolded protein accumulates in these cells.

19. "We found that *smedwi-1*(RNAi) animals express increased levels of the lysosomal gene *b-galactosidase*, which is a common marker of senescent cells"
This could be due to *smedwi-1*'s regulation of cathepsin cells which contain more lysosomal contents.

It is possible that cathepsin cells contain more lysosomes than other cell types. However, in the *smedwi-1*(RNAi) animals, cathepsin genes were generally decreased (Figure 3A), and thus a decrease in *b-galactosidase* would have been expected in that case. The detected increase in *b-galactosidase* cannot be explained by that.

20. "Based on our data, deregulation of RNA control can directly cause global age-associated defects such as reduced wound repair and accumulation of protein aggregates. This conclusion is unjustified. No direct data support this conclusion.

SMEDWI-1 is an RNA-binding protein, involved in RNA control, and we show the regulation of 7SL RNA and the SRP complex by this protein. We show that reduced wound repair and increased protein aggregation occur as a the result of loss of SMEDWI-1. Reduced wound repair and increased protein aggregation are defects that are commonly associated with aging. We therefore feel that our statement is therefore supported by our data. Nevertheless this statement was reworded in the context of the revision.

Reviewer 2

SUMMARY OF THE ADVANCE MADE IN THIS PAPER AND ITS POTENTIAL SIGNIFICANCE TO THE FIELD

In this study, the authors state that planarian worms knocked-down for SMEDWI-1 develop deficiencies in wound healing, which they interpret as a sign of ageing. Using these *Smedwi-1*(RNAi) animals as a model for studying aging, the authors observe that the extracellular collagen in the basal membrane is reduced, that the secreted epidermal structures are dysregulated and that the epidermal cells contain aggregates. At the molecular level, they identify a misregulation of transcripts in the late epidermis, namely a destabilization of the transcripts coding for secreted proteins, leading to a significant reduction in the number of rhabdoid epidermal structures. Finally, they propose that this secretion defect in *Smedwi-1*(RNAi) animals results from a misassembly of the Signal Recognition Particle (SRP) complex, with a progressive accumulation of 7SL RNA that becomes sequestered in large aggregates rather than being included in the mature SRP complex. The authors conclude that *Smedwi-1* is a key actor in essential cellular mechanisms such as SRP and proteostasis, necessary to support cellular functioning and to prevent aging.

This study provides novel and significant results on the role of *Smedwi* in maintaining epidermal homeostasis in planarians by controlling protein secretion. The results as they currently exist are convincing and the figures are generally clearly presented.

SUGGESTIONS TO AUTHORS

However, the authors' claims often go beyond the data they present:

- They show a convincing wound healing phenotype, but no data on whole body regeneration. Therefore, they should not write: "Depletion of SMEDWI-1 leads to defective regeneration".

We thank the reviewer for their appreciation of our work and their careful evaluation of the claims in our manuscript. We apologize for not making this more clear: The animals that do not close their wounds remain arrested in their regeneration, meaning that they will not regenerate into a intact planarian and will eventually die. We have now clarified this in the results.

- Similarly, in the absence of any characteristic sign of aging (decreased animal survival, decreased mitotic index of neoblasts, increased apoptosis, appearance of senescent cells, etc.), the link between the wound healing phenotype and the aging process has not been established in *Smedwi-1*(RNAi) animals. The authors cannot conclude their paper by writing: "In the absence of SMEDWI-1 the planarians become mere mortals, similar to most other animals, with similar age-related defects."

Therefore, either additional results must be added, or the presentation of the results and their discussion must be thoroughly revised before publication. Even in the absence of a reference to regeneration and aging, they will remain of great interest to cell biologists and developmental biologists.

We agree that there is an interesting discussion to be had on what constitutes a “sign of aging” . Although there is a suite of aging-associated phenotypes that has been found throughout multiple systems, each system presents with a slightly different subset, and some phenotypes are just not that consistent. For example, in some stem cell systems mitosis decreases with age - in others (such as haematopoietic stem cells) it increases, and in others it doesn't change. Cellular senescence is a process that looks different in every context in which it has been identified, and many of the described attributes of senescent cells only exist in vertebrates (e.g. SASP), and thus could never be detected in planaria. The main unifying phenotype of cellular senescence is the increase in b-galactosidase - which we do see in the *smedwi-1(RNAi)* animals. We also do detect increased fragility and reduced survival of tissue fragments, which we have now documented in Supplemental Figure S2. Further, we now analysed levels of apoptosis in homeostatic animals and found that the long-term *smedwi-1(RNAi)* animals show significantly increased levels of cell death even in the absence of wounding, now shown as Figure 6H-I.

Taken together, in the absence of SMEDWI-1, over the course of several months, the animals start to show several phenotypes that compromise organismal maintenance and are commonly found in aging in other systems. *Smedwi-1(RNAi)* animals don't show every phenotype associated with aging in the universe of model systems, and that could not reasonably be expected as these phenotypes frequently conflict with each other. *Smedwi-1(RNAi)* animals clearly show signs of proteostasis defects, protein aggregation, changes in ECM, increased levels of b-galactosidase, increased apoptosis, defects in wound healing, reduction of regenerative ability, increased fragility, and increased death - all of which are commonly associated with aging. Of note, these phenotypes are not independent: the increased death of tissue fragments is (most likely) caused by the defect in wound healing. However the same is true in naturally aging systems - the phenotypes can be interdependent and we do not have a complete understanding of the causalities.

Nevertheless, we understand the hesitation of the reviewer. Our statement that SMEDWI-1-less planarians become mortal was indeed posed in a somewhat provocative manner, and we now clarified that this is merely our hypothesis based on the data presented, rather than something that we would consider to be proven by our data.

A series of additional points need to be revised before publication:

- Regarding the efficiency of RNAi, what is the level of Smedwi-1 expression after 2 or 3 months of RNAi? Do the authors find a correlation between the intensity of the phenotype and the Smedwi-1 expression level?

For the level of SMEDWI-1 protein, please see Figure 2I. Knockdown of *smedwi-1* is very efficient and rapid. We have not detected a correlation between knockdown level and phenotype, although this would require evaluation of SMEDWI-1 level and regeneration outcome per individual animal, and we have not pursued this. We only periodically confirmed knockdown by pulling random animals from a cohort. We did notice that brief lapses of the RNAi (a break in feeding or accidental feeding with a defective batch of dsRNA) result in a reversion of the phenotype to the wildtype, suggesting that SMEDWI-1 has to remain close to fully suppressed to build up this phenotype.

- Why is the amputation level varying between experiments, i.e. anterior in Figure 1A and posterior

in Figure 1F? Are the phenotypes identical?

The main determinant for the phenotype is the size of the wound (in addition to the long-term *smedwi-1* knockdown of course). We tend to avoid amputations through the pharynx as this can cause complications in regeneration. The animal width is typically larger just anterior of the pharynx than posterior to the pharynx, and therefore we routinely performed the amputations at the anterior position. Measuring of neoblast migration to the wound site however would be complicated by the presence of the pharynx in such anterior amputations and thus for those experiments an amputation posterior to the pharynx was used.

- Failure to repair wounds may reveal an aging process as frequently observed in animals whose aging is well established, but it cannot be sufficient on its own to define an aging process. What is the organismal or cellular evidence that aging is actually at work in *Smedwi-1*(RNAi) animals?

The finding that over the course of several months the animals develop deregulated proteostasis, formation of protein aggregates, alterations in ECM, and upregulation of lysosomal genes, suggests that the planarian tissues are losing their optimal form over time in the absence of *SMEDWI-1*. Further, these phenotypes are commonly found during aging in other systems. For these reasons we referred to them as aging-related phenotypes in the context of this manuscript. We have now clarified the text in this regard. Whether this mimics natural aging in planarians, or rather reflects a loss of the ability to maintain organismal health long-term, is indeed unknown.

- Is the gradual loss in collagen observed in homeostatic animals, independently of any amputation? This is not clear in Figure 2E, 2I.

This is indeed in homeostatic animals. We have now clarified this in the legends.

- The heading (p9): "*Smedwi-1*(RNAi) animals misregulate transcripts of epidermal precursor cells" is confusing as in Figure 3A, transcripts specific to late epidermis are strongly down-regulated whereas transcripts from other cell types, including epidermal precursors, are not. Similarly, the legend of Figure 3A does not seem to reflect the results shown in this panel.

We apologize for the confusing naming. The naming of the gene sets is derived from a previous single cell study. The "late epidermal" category however reflects late epidermal precursors. These are still not mature epidermal cells. The developmental order is: early epidermis - late epidermis - mature epidermis. We have now clarified this by renaming the "early epidermis" and "late epidermis" to "early epidermal precursors" and "late epidermal precursors".

- In Figure 3E, the legend is either unclear or not appropriate as no epidermal precursor cells are shown

Agat-3 is a marker for the late epidermal precursors. We have altered the label to clarify this.

- In Figure 3D and 3F, it would be clearer to write directly on the figure the meaning of the color shadings

We have adjusted this in the figure.

- In Figure 4A, more details on the histological view would be welcome; for example, what are the purple structures above the Basement Membrane?

These are the epidermal cells, and further distal are secreted rhabdoids and mucus. We have added this information in the figure.

- In Figure 4B, what are the large electron-light structures that are visible in the upper right panel? what does it mean "Rh?" in the lower right panel?

The light structures are places where rhabdoids normally might have been located. We cannot

determine whether they were missing in the *smedwi-1(RNAi)* animals, or whether due to altered composition they were washed out during the fixation process. Rh indicates Rhabdoid, as indicated in the legend.

- The authors cite the extensive review on planarians as a model for understanding biological ageing (Sahu et al., 2017), but I would recommend that they also refer to the few experimental studies available on this question, e.g. Perrigue et al., 2015; Gambino et al., 2020; Deere et al., 2024

We thank the reviewer for suggesting these papers. We have added these references.

- This study describes cellular and molecular changes that take place exclusively in epidermal cells, therefore the authors should clarify in the discussion the two distinct roles played by Smedwi in planarians, in neoblasts on the one hand, and in the epidermal cells on the other hand.

This is an interesting conundrum. The SMEDWI-1 protein is detected in the neoblasts and the early post-mitotic cells (such as the early epidermal progenitors). By the time the physiological changes to the cells are visible (late epidermal progenitors), SMEDWI-1 protein is no longer detected (although of course it is possible that SMEDWI-1 protein remains present at levels that are below our detection limits). It thus appears that the defect to the non-coding RNA that already arises in the neoblasts and early differentiating cells, results in a collapse in proteostasis in highly secretory cells that derive from these *smedwi-1(RNAi)* neoblasts. SMEDWI-1 protein as such thus does not play a role directly in the epidermal cells, but it plays a role in setting up the stem cells in such a manner that they can become functional epidermal cells once they differentiate. This is not to mean that other cell lineages are not affected by the loss of SMEDWI-1 (and indeed we detect milder effects on transcripts encoding secreted proteins in muscle and in cathepsin cells) - however the defect probably shows primarily in the cells that have high secretory activity as this will push the secretory machinery to its limits.

- A summary scheme clarifying the link between the concomitant reduction in rhabdoid epidermal structures and the appearance of aggregates in late epidermal cells would be useful.

We thank the reviewer for this suggestion, and we have added a schematic as Figure 7B to clarify this.

- How do the authors explain that *Smedwi-1(RNAi)* animals do not exhibit any organismal phenotype?

Most studies had only investigated short-term knockdown of *smedwi-1*, up to 3 weeks. In this time window the RNA defect is only starting to build, and it doesn't reach the level of dysfunction required to disrupt 7SL and protein secretion. It is most likely for this reason that investigators so far had not noticed the phenotypes described in this study.

It is important to note that even at the extended periods of *smedwi-1* knockdown as used in this study, the proteostasis defect is only detected in highly secretory cells that probably are particularly sensitive to mild misregulation of protein secretion. The defects in other cell types may take even longer to develop, and/or may only show when the cells are challenged. In this context it is interesting to note that while we did not identify any major phenotype in homeostatic *smedwi-1(RNAi)* animals, they do tend to become more sickly than control RNAi animals over time. Asexual animals maintained on *smedwi-1* RNAi generate fewer fission fragments that regenerate successfully, and as a result their colonies expand more slowly. We have now added supplementary text and illustration in Supplementary Figure S2 to document this.

Reviewer 3

In this paper, the authors have investigated more fully the phenotype of *smedwi-1(RNAi)* animals, a pleiotropic grouping of phenotypes that include altered RNA abundance that they originally uncovered in a prior paper (Allikka Parambil, et al). They investigate decreased wound healing in long-term *smedwi-1(RNAi)* animals, determining that stem cell abundance is unaffected, but that late epidermal progenitors are impacted. They also show that wound response genes are elevated

long after amputation, suggesting failure to resolve the wound response. The authors also detect changes in cellular structures, including putative extracellular matrix and rhabdoids, before attempting to connect these ideas through the discovery that the non-coding RNA associated with the signal recognition particle is somewhat upregulated after *smewi-1*(RNAi). Finally, the group uses Proteostat to test for possible protein aggregation in *smewi-1*(RNAi) animals. Taken together, the group concludes that *smewi-1*(RNAi) induces a suite of phenotypes overlapping with those seen in aging in other animals but not yet observed in planarians. Some of the ideas in this paper are really strong— understanding how *Smedwi-1* regulates non-coding transcripts is very interesting and a potential understudied aspect of PIWI biology. Understanding why long-lived, regenerative species like planarians do not normally show signs of aging is also a fascinating research topic. However, enthusiasm for some of the goals of the research project is damped by a few considerable technical and interpretation issues. Thus, this paper does not yet meet the standards of rigor of the journal *Development* at this time. Major and minor concerns follow.

Major concerns:

1) The conclusion that the suite of phenotypes seen in *smewi-1*(RNAi) animals represents "aging" is undersupported, particularly given some of the technical concerns with individual experiments (detailed below). One can imagine that a large number of perturbations in core RNA processing machinery might lead to a suite of outcomes that includes proteostasis issues, altered mRNA and protein production, and so on. At a minimum, the arguments laid out below would need to be strengthened to confirm that the phenotypes shown are a fair approximation of aging. Otherwise, the aging connection could be softened and moved to the discussion to mitigate this concern.

We appreciate this concern and we have similarly debated this at length. There is an interesting philosophical discussion to be had on what really constitutes aging, other than the progressive loss of organismal function over time - because this is clearly what we observe: after the elimination of *SMEDWI-1* is complete, the animals show a gradual loss of organismal function over the course of several months, and they show phenotypes that in other systems are commonly associated with aging.

It is probably true that a large number of molecular perturbations lead to compromised organismal function. What sets the phenotype of *smewi-1*(RNAi) apart is the following: 1. The molecular perturbation itself is fast (within two weeks *SMEDWI-1* protein is no longer detectable), but the phenotype takes several months to develop. Planarian tissues turn over every 3 weeks, and therefore a phenotype delay of several months is very uncommon and suggests progressive deterioration of organismal function. 2. The phenotypes observed are a subset of progressive degenerative phenotypes that are commonly observed during aging in other systems. This is not true for every molecular perturbation, and the occurrence of this aggregate of phenotypes makes this at least remarkable. 3. The phenotypes are associated with the loss of a factor that in other animals is present in germ cells and early embryogenesis but is lost afterwards - when aging occurs. Also, this factor remains present in the stem cells in several systems that show negligible senescence.

The question remains whether this should be called aging in planarians, and based on the comments of the reviewers we think that this is an unnecessarily divisive concept that we can limit to the discussion.

2) The 6G10 antibody was used to label "collagen" in Fig. 2 and throughout, but this antibody, developed by Ross and colleagues, is described in their paper as labeling muscles. It has also been used in the field by other researchers as a muscle marker. Therefore, the "collagen fibers" described in Fig. 2H are more likely to be "muscle fibers." It is also unclear what protein is being visualized in the western blots since to my knowledge the identity of the antigen that this antibody binds has not yet been determined. If the group has evidence that this antibody binds collagen (e.g. RNAi targeting a collagen gene reduces staining and western blot signal), that should be provided. Many of the conclusions drawn using this antibody should be revisited so that we can understand if it is the muscle that is affected or the ECM. If, indeed, the muscle is affected through the staining, then other results, like the differential expression of muscle genes might become more central to the work. Alternatively, There is a collagen IV antibody used in Dubey, et al, which might be useful for testing ECM changes.

We thank the reviewer for bringing this up, as well as suggesting potential strategies to resolve this

issue.

The reason we proposed that the epitope represents a collagen is that the signal is decreased upon treatment of samples with collagenase. The band on Western is detected both in protein extracts from whole animals, and in protein samples from cells dissociated in calcium-free buffer. However, when the dissociated cells are treated with collagenase, the signal is strongly reduced, whereas there is no effect on a non-collagen protein such as Tubulin or Myosin. This indicates that the antibody recognizes an extracellular protein (it is accessible in dissociated cells) that is cleaved by collagenase. We appreciate that extracellular proteins other than collagens may be cleaved by collagenase, and we have tried to narrow down the epitope by testing the signal in RNAi samples that have knocked down fibrillar collagens. We found that RNAi of *colf-8*, *colf-9*, and *colf-10* resulted in a reduction of the 6G10 signal on Western blot, and RNAi of *colf-9* also showed disruption of the staining in immunofluorescence. Based on this data, we propose that a collagen is the most likely candidate for the epitope of this antibody, and this fits well with the reported staining of muscle fibers, which are the major source of collagens, in fixed animals.

We do not interpret the reduction in 6G10 signal as a reduction in muscle because we found that the number of muscle cells as detected by FISH for collagen-2 was not reduced in *smewi-1(RNAi)* animals (Figure 3E), and that the levels of non-secreted muscle transcripts were also not significantly altered (Suppl. Figure 3E), arguing against a general loss of muscle cells. We now also used the well-characterized Myosin antibody TMUS (Cebria 1997) to demonstrate the continued presence of muscle in *smewi-1(RNAi)* animals by immunofluorescence. We have included these data in Figure 3E. Unfortunately as TMUS and 6G10 are both mouse antibodies, we were not able to perform double labelling of these epitopes.

The reduction of the basement membrane is clearly visible from the H&E staining and the EM images, and we therefore did not pursue this further by Collagen IV staining.

3) The primary impact of *smewi-1(RNAi)* occurring in epidermal and epidermal progenitor cells made me wonder whether equinox mRNA is impacted after *smewi-1(RNAi)*. Equinox encodes a

secreted protein from the epidermis and is absolutely required for blastema formation (Scimone, et al 2022 Nature Communications). Due to similarities in the long-term *smewi-1*(RNAi) phenotype and the *equinox*(RNAi) phenotype, it would be helpful to understand whether there is a direct connection between the two genes.

We thank the reviewer for this interesting connection. From the original publication, it appears that loss of *equinox* leads to the inability to maintain wound-induced gene expression, whereas we find an inability to downregulate wound-induced gene expression. Nevertheless we tested the induction of *equinox* at 16h post wounding (based on original study this was the timepoint of maximum induction), as shown in the graph below.

We find that the wound-induced upregulation of *equinox* is more subtle than that of other typical wound-induced genes such as *egr-2* and *runt-1*. *Equinox* is still upregulated in *smewi-1*(RNAi) wounds, but this is weaker ($P < 0.0001$) than in the controls. It is possible that the mRNA level is reduced due to the fact that it encodes a secreted protein and is not correctly processed. Based on the data we feel comfortable stating that some transcriptional induction of *equinox* still occurs in the absence of SMEDWI-1, and given that we find the opposite phenotype compared to the inhibition of *equinox* it appears that the amount of protein produced is sufficient to support the sustained expression of wound-induced genes.

4) There are some inconsistencies between the size of "rhabdoids" seen in EM images (~ 2-3 μm long) and the stained mucus trail images (10 μm +) and in stained images (20-30 μm). Further, the Rh structures in Fig. 4B do not look a lot like rhabdites in other EM images (Hayes 2017 is a good reference). This left me wondering if the structures visualized are all "rhabdoids." Because these structures are not well characterized structurally or functionally in the literature, more evidence to support the consistent identity of these structures would help strengthen the argument that these structures are changed after *smewi-1*(RNAi). Are there other papers documenting that aniline blue or Lens Culinaris Agglutinin stain rhabdites/rhabdoids? Did you try Cupromeronic Blue (Hayes 2017)? Do they stain with microtubule antibodies as has been suggested in Lentz (1967) and Kornakova (2013)? Do rhabdoids have known function in regeneration or other "aging" phenotypes? For what it's worth, I do think they are really interesting and worthy of study!

This is an interesting issue and we have now added further details in the supplement to explain this better. We share the view that the rhabdoids/rhabdites that are referred to in literature are probably not one singular structure, as the observations are not fully consistent. There probably are several distinct secreted structures that differ in their composition, and this (in addition to the rapid disintegration of the structures) likely explains the discrepancies between the reported observations. To not add to the confusion, we decided to refer to the structures we observed as "rhabdoids", intending this to mean rhabdite-like structures. This naming was previously applied in Smith III 1982 ("The Morphology of Turbellarian Rhabdites: Phylogenetic Implications").

Based on our experiments, the rhabdoids are highly hygroscopic and inflate immediately upon secretion. In the mucus trails or in high salt solution they remain recognizable for several minutes, but eventually they turn into smears of mucus that merge with the remains of other rhabdoids and do not retain any recognizable shape. If put in contact with buffer of lower osmolarity they

disappear almost instantly. This was also observed in (Hayes 2017). The different sizes of the structures reflect the different stages of this rapid inflation process. Due to the rapid dissolution of the rhabdoids, they are difficult to stain or fix once they have been secreted. The staining strategies we used relied on reagents that stained the rhabdites instantly allowing us to observe the stained structures within seconds. Aniline blue was inspired by the toluidine blue staining used by Hayes and alcian blue used by Wrona (Hydrobiologia, 1986); all are dyes that bind negatively charged structures. LCA has not been previously described to stain rhabdoids, but was one of the other rapid stains that we tested and that provided specific and reproducible labelling. We found this labelling particularly informative because it was clearly distinct between control rhabdoids and *smedwi-1(RNAi)* rhabdoids. We did not try tubulin antibody staining as we did not find satisfactory ways to fix these structures for immunofluorescence. However tubulins have been found in a proteomic analysis of planarian mucus (Bocchinfuso 2012), and thus they are likely to be present as part of the secreted structures.

To connect the various rhabdoid structures that we observed, we present the following. We used 3 methods to visualize the structures: 1. EM showing the rhabdoids within the epidermis; 2. Salt extraction and staining of the excreted rhabdoids by aniline blue or LCA; 3. Staining of mucus trails by aniline blue showing remaining rhabdites in the natural secretions of the animals. The aniline blue staining of mucus trails and salt excreted structures strongly suggests that these are the same structures: they look very similar, and after salt extraction few additional rhabdoids are secreted in the trails. The structures identified by salt extraction take different shapes over time, and these changes are observed both by aniline blue and by LCA staining as now shown In Supplementary Figure S4. The LCA staining is most informative on the slightly inflated structures, and that is what we had shown in our manuscript, but initially the LCA-stained structures have the same size and shape as the ones that we showed by aniline staining. Further, we can detect inflated structures by aniline staining as well and they arise at a similar time as the inflated structures seen in the LCA staining. Finally we tried to connect these secreted structures to structures observed in the EM. To accomplish this we now added EM images of the epidermis of a salt-extracted animal in Supplemental Figure S4. This shows the loss of the larger rhabdoid structures from the epidermal cells after this treatment.

We could not convincingly identify changes in the various types of smaller epidermal inclusions in response to salt treatment. We hypothesize that these smaller epidermal inclusions are precursor stages to the larger rhabdoids and are not sufficiently mature to be excreted in response to the salt, but alternatively they could be independently formed secreted structures. We do detect a reduction in these smaller inclusions in the *smewi-1(RNAi)* animals, but as we cannot be certain of the identity of these smaller structures we decided to remove the images showing and labelling them from the main figure and retain only the EM images of the larger rhabdoids that resemble previously published rhabdoid images.

We agree that the planarian rhabdoids are intriguing, and that there is probably a whole lot of exciting biology that can be discovered by investigating them in more detail. There is only a handful of studies spread over various species describing these structures and little is known about their biogenesis, their composition, or their biological effects. We however feel that further characterization would require a dedicated effort and that it is not feasible to solve this in the context of this current manuscript.

5) The potential perturbation of 7SL in *smewi-1(RNAi)* was interesting, but there were a few issues with the size fractionation experiment. The UV graph at the top should have axes labeled. It is not clear how the group determined kDa assignments for the X axis when typically fractions are noted on this axis. It would be helpful to understand how the group assigned kDa values in this experiment given that flow-through rate of protein complexes can be impacted by both mass and shape of a complex. The UV graph also lacks some of the peaks seen in the Supplemental Figure so I am not sure how the graph in G was generated. More evidence would also be required to show that 7SL is present in aggregates rather than in higher-molecular weight, physiologically-relevant complexes.

We thank the reviewer for requesting these clarifications. The UV profile shown at the top of Figure 5G shows the UV absorbance plotted against the retention volume (which results in the fractionation). The UV absorbance as recorded by the flow cell is given in unitless "absorption

unit”, which is set to 0 at the absorbance of the eluent. The scale is now indicated in the figure. As the elution volume is not very informative, the location of size markers are indicated on the x-axis instead. The UV profile is lined up to correspond to the fractions on the gel.

Size fractionation columns are typically calibrated by running a set of purified proteins of known molecular weight over the column and noting their retention time. In this case we calibrated the column with Thyroglobulin (670kD), Apoferritin (430kD), Beta-amylase (200kD), Bovine Serum Albumin (66kD), and to mark the total column volume acetone (0kD). These profiles are highly reproducible and were repeated several times throughout the runs. We have now added this information to the Methods.

Figure 5G shows the first 13ml (52 fractions) of the elution. At that point all complexes larger than 20kD should have come off the column. Peaks that occur after that point in the Supplemental figure are not part of the section shown in Figure 5G. The remaining absorption peaks probably represent LMW compounds and metabolites.

It is correct that the retention time of proteins depends on their conformation in addition to their molecular weight. Size exclusion chromatography routinely assumes that all proteins and all protein complexes are globular - even though this is a simplification. Fibrous proteins would likely be retained longer and thus appear to have a lower molecular weight than their molecular composition suggests.

We did not intend to make a distinction between aggregates and high molecular weight complexes. It is clear from the gel that in the *smewi-1(RNAi)* sample, the majority of 7SL RNA is not present in the fractions that correspond to the mature SRP complex, but rather are in a much higher molecular weight fraction. We have now clarified the text to reflect that this could represent either aggregates or HMW complexes.

6) Proteostat has, to my knowledge, not yet been validated in planarian cells. It would be helpful to see whether manipulations like MG132 treatment or chaperone knockdown result in increased proteostat staining, to confirm the validity of this tool in planarians. I was especially surprised to see a nuclear or perinuclear staining in Fig. A-B, given that most examples in the literature show Proteostat excluded from the nucleus. Another option here is to find a second strategy to support the conclusion that proteostasis is deregulated.

We are also not aware of previous descriptions of Proteostat staining in planarians, and we agree that careful validation of new methods is important. We have performed RNAi against *hsp70* to increase misfolded protein, and have used MG132 to reduce protein turnover by the proteasome. Both treatments resulted in increased Proteostat staining in the epidermal and subepidermal regions. This data is now shown as Supplemental Figure S6B.

With regard to the localization of the staining, this is indeed not nuclear. The image in Figure 6D (previously 6B) is a 2D projection of a z-stack, and therefore some foci appear to overlap the nucleus whereas they are in fact in a different plane, in a different region of the cell. We have now added this note to the figure legend.

Additionally we now include data showing a global increase in ubiquitinated protein in *smewi-1(RNAi)* animals, which serves as an additional indication that proteostasis is deregulated under these conditions (Supplementary Figure S6A).

Minor concerns:

1) All antibodies that have not been validated in planarians before should be validated by RNAi and western blot experiments (e.g. SRP54) to verify specificity. We have now added this as Supplemental Figure S5E.

2) It is not always clear what statistical test was used for each experiment (e.g. Fig. 1E, right) and some statistical analyses may need to be reevaluated. N should also be provided for all experiments (e.g. Fig. 1F), either in the image or figure legend.

We have added this information.

3) The white/DAPI staining of the *smewi-1(RNAi)* animal in Fig. 1F (and even clearer in the supplement) has some large white areas that don't look nuclear. Is this a real phenotype present in

all animals in this experiment?? If so, what is this staining?

These structures are indeed present in most of the animals, and they appear to be the sperm ducts. These structures are typically resorbed during regeneration, but in the *smewi-1(RNAi)* animals this is delayed or abrogated, possibly due to the same lack of signalling that causes the diminished neoblast accumulation at the wound site. We have clarified this in the supplementary figure legend.

4) Can the authors please clarify why they called structures rhabdoids rather than rhabdites?

The literature on planarian rhabdites is quite sparse. There are several beautiful older EM studies that have used a variety of species and found different organizations of these structures in different planarians. In *Schmidtea mediterranea* there appear to be several distinct structures that may all be stages of rhabdites, or may be distinct rod-shaped secreted rhabdite-related structures. We chose the term “rhabdoid” to mean rhabdite-like structure as we were not able to identify the observed structures more precisely.

5) Does *smewi-1(RNAi)* cause eventual lethality? Given the emphasis on aging, this might be helpful to document.

We do not detect overt lethality among uninjured homeostatic *smewi-1(RNAi)* animals. This is however also not a phenotype that is commonly observed in planarians unless in cases of major stem cell failure. Loss of animals typically occurs by gradual shrinking to the point of no return, or defective regeneration followed by lysis of shrinkage. In *smewi-1(RNAi)* asexual cultures, the various non-regenerating fragments that are formed over the course of the animal culture die in this manner, and fragments with large wound surfaces (exemplified by the parasagittal amputations now shown in Figure S1C) that survive and regenerate fine when derived from control RNAi animals, lyse rapidly, often leaving little recognizable trace. In agreement with this we detect that even in the absence of intentional amputations, asexual colonies on *smewi-1* dsRNA expand notably less than colonies on control dsRNA. Further, we detected that homeostatic *smewi-1(RNAi)* animals tend to become more sickly than control RNAi animals over time, and that regenerative malformations can be detected in around 15% of long-term *smewi-1(RNAi)* animals whereas they are very rare (<1%) in controls. Further, in the absence of antibiotics, lysis of cut fragments and development of abnormalities is exaggerated in the *smewi-1(RNAi)* animals, whereas no changes are observed in the controls, suggesting that the *smewi-1(RNAi)* animals are more fragile in face of microbial challenges. We have now added brief documentation of these phenotypes in Supplementary Figure S2 and added Supplementary text.

Curiosity questions:

1) If the mature mRNA degradation in Fig. 3H is via RAPP, is it dependent on a planarian homolog of Ago2?

This is a really interesting question. We have not looked into this specifically, but as our lab is generally interested in small RNA regulation and Argonautes/PIWIs, we may well run into this in the future!

Second decision letter

MS ID#: dev.204762R1

MS TITLE: Enhanced RNA quality control maintains long-term regenerative ability in planarians

AUTHORS: Michael Zelko, Danyan Li, Andrew Verdesca, Sudheesh Allikka Parambil, Axel Poulet, Kaspar Mazeika, Krishnakali Dasgupta and Josien van Wolfswinkel

Dear Dr van Wolfswinkel,

I am happy to tell you that your manuscript has been accepted for publication in Development, pending our standard publication integrity checks. It was accepted on 04 Sep 2025. Where referee reports on this version are available, they are appended below. As you will see, one reviewer still raised some concerns. Editorially, I disagreed with some of the substantial points made by this reviewer, but I recommend that you could consider making some edits in the writing of the manuscript to ensure other readers have a clearer perspective on these points.

Reviewer 1

Advance summary and potential significance to field

The authors' efforts in the response to comments are greatly appreciated. Even though "aging" is no longer used in the abstract and results, the key message the authors deliver from title, abstract, summary statement, results, to discussion remains the same: loss of SMEDWI-1 leads to progressive loss of organismal vigor, in other words, aging. Interpretation of the data in the context of "long-term" organismal vigor, or "aging" remains problematic.

Responses to some of the earlier concerns were not sufficient. In addition, the authors' new responses revealed concerns that make the conclusions of current study more questionable. For example, the data is collected from animals maintained continuously in gentamycin for 2-6 months.

When essential genes are lost, it is expected that "life-long" maintenance of an animal's health and regenerative abilities will be lost. To support the statement that "life-long RNA control mediated by a stem cell PIWI protein" regulate planarian longevity, health and regenerative abilities, the authors need to distinguish between "essential functions" and "long-term functions", which is not provided in the study.

The authors made the effort to delay the death of animals upon loss of SMEDWI-1, by supplementing gentamycin continuously. This method delayed the death of Smedwi-1 KD animals from 3-4 weeks to 2-3 months (sup Fig.2C-D, and other texts). In the gentamycin conditions, the authors found phenotypes became more severe in 3 months, compared in 2 months (Fig. 1A-B). This became the foundation of the authors' study to support their conclusion that "life-long" RNA control mediated by SMEDWI-1 regulates planarian's longevity and life-long regenerative abilities.

While it is likely SMEDWI-1 has important functions in life-long health and regenerative abilities of the planarians, especially as an essential gene, experimental designs in the current study does not justify such a conclusion.

1. 2-3 months of time is still a short period of time in a planarian's life. Delayed death with gentamycin treatment is not sufficient to support "life-long" regulation.
2. SMEDWI-1 is an essential gene. This is also supported by death in 3-4 weeks without gentamycin. "The fragility of the smedwi-1 animals is such that without Gentamycin the long-term knockdown experiments would not be possible as the majority of the animals would be lost or disorganized".
3. Gentamycin or antibiotics in general can have complex effect on an animal's physiology. One famous example is the use of neomycin to induce hair cell death in zebrafish.
<https://link.springer.com/article/10.1007/s10162-002-3022-x>

Other examples of stem cells:

"Toxic Effects of Gentamicin on Marrow-derived Human Mesenchymal Stem Cells"

https://journals.lww.com/clinorthop/abstract/2006/11000/toxic_effects_of_gentamicin_on_marro_w_derived.45.aspx

"Effect of Gentamicin on Growth and Differentiation of Human Mesenchymal Stem Cells"

https://www.jstage.jst.go.jp/article/tox/21/1/21_1_61/_article/-char/ja/

4. There is likely a strong interaction between Smedwi-1 perturbation and gentamycin. Without gentamycin, smedwi-1 RNAi animals will be lost in 4 weeks. Hence, gentamycin has to play an important role in the reported data from 3-month-long experiments. The authors acknowledged that gentamycin inhibits microbiota. The presence of microbiota and innate immune response are critical components of life-long regulations of animal physiology. The authors carefully acknowledged that the control animals do not show a phenotype as severe as the RNAi animals. In other words, the control animals under gentamycin did develop similar phenotypes. For example, defects in wound healing and regeneration as shown in Fig 1. A-C. Smedwi-1 RNAi enhanced the phenotypes especially during prolonged gentamycin treatment. Phenotypes presented in the current study can be a combined effect of long-term gentamycin treatment and smedwi-1 RNAi.

Examples of insufficient response to previous concerns:

Fig. 2A: increase of h2b, RPSAP58, DUT, TOPBP1 in neoblasts after smedwi-1 RNAi. This suggests loss of SMEDWI-1 has an impact on neoblasts, different from what the authors wanted to conclude.

Fig. 2E: epidermis is much thicker in smedwi-1 RNAi animals than in control animals. More collagen in smedwi-1 RNAi. Yet, Fig. 2F: less collagen at wound surface.

"Enhanced RNA quality control" in the title needs to be supported by gain of function experiments. The current study is a loss of function study.

Reviewer 2

Advance summary and potential significance to field

The article has been thoroughly revised by the authors, and my comments, questions, and criticisms have been adequately addressed. In my opinion, this article is now ready for publication as it provides new and important insights into the impact of Piwi on the regeneration process and resistance to aging processes, particularly with regard to RNA stability, protein secretion, and aggregate formation.

Comments for the author

well-done!

Reviewer 3

Advance summary and potential significance to field

Proteins that bind and regulate RNAs are common in stem cells and are a feature of planarian stem cells, also known as neoblasts, in particular. Smedwi-1 has long been the go-to marker for planarian stem cells, yet its cellular functions and roles in stem cell maintenance have been slow to emerge. The authors complete a nice dive into Smedwi-1 function, showing that it is key in regulating the non-coding RNA that is critical for the Signal Recognition Particle. In the absence of Smedwi-1, maladaptive accumulation of aberrant ncRNA leads to a dysfunctional SRP system and eventual proteostasis crisis. Eventually, the cellular integrity of the animal declines, which is evident is a reduction of regenerative success. This work provides a nice blend of biochemistry and organismal biology to understand the function of key, conserved RNA regulatory mechanisms.

Comments for the author

The authors have done a nice job of addressing the concerns of the reviewers through both writing and experimental changes. I think that the decision to move aging into a more speculative portion of the paper was a good decision and would like to suggest to the authors that short-lived, seasonal

planarians (like *Procotyla*) might be helpful to use for defining true aging phenotypes for planarians in the future. That would allow them to come back to interesting phenotypes like the ones shown here and more unambiguously link them to aging.